# Discovering a mitochondrion-localized BAHD acyltransferase involved in calystegine biosynthesis and engineering the production of 3β-tigloyloxytropane

Junlan Zeng [1], Xiaoqiang Liu[1], Zhaoyue Dong[2], Fangyuan Zhang[1], Fei Qiu[1], Mingyu Zhong[1], Tengfei Zhao[1], Chunxian Yang[1], Lingjiang Zeng[1], Xiaozhong Lan[3], Hongbo Zhang[4], Junhui Zhou[5], Min Chen[2], Kexuan Tang[1,6] & Zhihua Liao [1] ✉

Solanaceous plants produce tropane alkaloids (TAs) via esterification of 3α- and 3β-tropanol. Although littorine synthase is revealed to be responsible for 3α-tropanol esterification that leads to hyoscyamine biosynthesis, the genes associated with 3β-tropanol esterification are unknown. Here, we report that a BAHD acyltransferase from *Atropa belladonna*, 3β-tigloyloxytropane synthase (TS), catalyzes 3β-tropanol and tigloyl-CoA to form 3β-tigloyloxytropane, the key intermediate in calystegine biosynthesis and a potential drug for treating neurodegenerative disease. Unlike other cytosolic-localized BAHD acyltransferases, TS is localized to mitochondria. The catalytic mechanism of TS is revealed through molecular docking and site-directed mutagenesis. Subsequently, 3β-tigloyloxytropane is synthesized in tobacco. A bacterial CoA ligase (PcICS) is found to synthesize tigloyl-CoA, an acyl donor for 3β-tigloyloxytropane biosynthesis. By expressing TS mutant and PcICS, engineered *Escherichia coli* synthesizes 3β-tigloyloxytropane from tiglic acid and 3β-tropanol. This study helps to characterize the enzymology and chemodiversity of TAs and provides an approach for producing 3β-tigloyloxytropane.

Plant secondary metabolites have been utilized as medicines by humans for thousands of years. Tropane alkaloids (TAs) are a class of secondary metabolites that are characterized by an 8-azabicyclo [3.2.1] octane core skeleton that contains a cycloheptane ring with a nitrogen bridge and are commonly referred to as a tropane moiety or tropane core[1]. More than 300 TAs have been identified from the *Solanaceae*, *Convolvulaceae*, *Rhizophoraceae*, *Erythroxylaceae*, and other families[2]. These compounds include the anticholinergic pharmaceuticals, hyoscyamine and scopolamine, which are synthesized by select genera of the *Solanaceae* family, and the narcotic cocaine, which is synthesized by the *Erythroxylaceae*[3].

Recent studies on *Atropa belladonna*, a species belonging to the *Solanaceae*, have comprehensively revealed the biosynthetic pathway

[1]Integrative Science Center of Germplasm Creation in Western China (CHONGQING) Science City, State Key Laboratory of Resource Insects, SWU-TAAHC Medicinal Plant Joint R&D Centre, School of Life Sciences, Southwest University, Chongqing 400715, China. [2]College of Pharmaceutical Sciences, Southwest University, Chongqing 400715, China. [3]TAAHC-SWU Medicinal Plant Joint R&D Centre, The Provincial and Ministerial Co-founded Collaborative Innovation Center for R&D in Xizang Characteristic Agricultural and Animal Husbandry Resources, Xizang Agricultural and Animal Husbandry College, Nyingchi 860000, China. [4]Key Laboratory of Synthetic Biology of Ministry of Agriculture and Rural Affairs, Tobacco Research Institute of Chinese Academy of Agricultural Sciences, Qingdao 266101, China. [5]State Key Laboratory of Dao-di Herbs, National Resource Center for Chinese Materia Medica, China Academy of Chinese Medical Sciences, Beijing 100700, China. [6]Fudan-SJTU-Nottingham Plant Biotechnology R&D Center, School of Agriculture and Biology, Shanghai Jiao Tong University, Shanghai 200240, China. ✉e-mail: zhliao@swu.edu.cn

of hyoscyamine and scopolamine[4,5]. The tropane moiety of hyoscyamine and scopolamine is contributed by 3α-tropanol (also named tropine), which is biosynthesized by tropinone reductase I (TRI) using tropinone as a substrate[6]. In addition, tropinone can be reduced to 3β-tropanol (also named pseudotropine) by tropinone reductase II (TRII) (Fig. 1). Calystegines, 3β-tropanol-derived TAs, are widely distributed in the plant kingdom and include several important vegetables in the *Solanaceae*, such as tomatoes, potatoes, eggplants, and hot peppers[6–9]. Calystegines strongly inhibit glycosidase, might cause problems with nutrient absorption, and can be used in therapies for metabolic disorders[10–13]. Despite being minor components in TA-producing plants, some 3β-tropanol esters are pharmaceutically important[14,15]. For example, 3β-tigloyloxytropane, otherwise known as tigloidine or tropigline, is a substitute for atropine in the treatment of neurodegenerative disease; compared to atropine, 3β-tigloyloxytropane causes fewer side effects, including dry throat, mydriasis, and headache, than atropine[16–18]. In addition, 3β-benzoyloxytropane, otherwise known as tropacocaine, is a potential ophthalmic and spinal anesthetic[19].

Comprehensive knowledge on the biosynthetic pathways of hyoscyamine, scopolamine, and cocaine has greatly facilitated their production in engineered plants and microbes[5,20–22]. However, knowledge on the biosynthesis of 3β-tropanol derivatives is limited. Therefore, the efficient production of valuable 3β-tropanol esters, such as 3β-tigloyloxytropane and 3β-benzoyloxytropane, using metabolic engineering or synthetic biology approaches is also limited. Structurally, it was postulated that calystegines can be directly demethylated and hydroxylated from 3β-tropanol[23–25]. Nevertheless, recent work challenges this long-standing speculation and propose the following roadmap: 3β-Tropanol is esterified into 3β-tigloyloxytropane (the key intermediate for the biosynthesis of calystegines), which undergoes demethylation and hydroxylation, followed by hydrolytic reactions, resulting in the production of calystegines[26]. Although Robins and colleagues reported that catalytic activities associated with 3β-tropanol esterification occurred in *Datura stramonium* and *A. belladonna* in the 1990s[27,28], the genes that encode acyltransferases responsible for the formation of 3β-tropanol esters remain to be elucidated.

Esterification is among the most important chemical modifications of small molecules and involves the addition of an acyl moiety to produce esters and amides. Two enzyme families, serine carboxypeptidase-like acyltransferase (SCPL-AT) and BAHD acyltransferase (BAHD-AT), function as major acyltransferases that are involved in metabolite esterification in plants[29]. SCPL-ATs usually employ 1-*O*-β-glucose as an acyl donors[29]. For instance, in the biosynthesis of hyoscyamine and scopolamine, the SCPL-AT littorine synthase (LS), catalyzes the condensation of 3α-tropanol and phenyllactylglucose via esterification to generate littorine[30] (Fig. 1). Interestingly, a recent study indicated that chicoric acid synthase (EpCAS), an SCPL-AT from *Echinacea purpurea*, uses chlorogenic acid as an acyl donor[31]. However, BAHD-ATs normally use acyl-CoA thioesters. For instance, *Erythroxylum coca* cocaine synthase (EcCS), a member of the BAHD-AT family, is responsible for the biosynthesis of cocaine (a TA produced by the coca tree) through initiating esterification between methylecgonine (an intermediate with a tropane moiety) and benzoic acid[32].

SCPL-ATs and BAHD-ATs resulted from independent evolution[33]. One of the fascinating aspects of metabolism is complex subcellular compartmentation. Many enzymes involved in secondary metabolite biosynthesis are located in the cytosol and multiple organelles. SCPL-ATs are usually localized in the vacuole, and this localization is essential for their function[5,20]. Generally, BAHD-ATs do not possess a localization signal and occur freely in the cytosol[29]. The enzymes involved in the biosynthesis of hyoscyamine and scopolamine are distributed throughout the cytosol, vacuole, and endoplasmic reticulum[5]. Interestingly, although mitochondria are the center of nitrogen metabolism and TAs are nitrogenous secondary metabolites[34], no mitochondrial localization of enzymes involved in TA synthesis has been reported.

In this study, we functionally identified a mitochondrion-localized BAHD acyltransferase from *A. belladonna*, 3β-tigloyloxytropane synthase (TS). This enzyme catalyzes 3β-tropanol and tigloyl-CoA to produce 3β-tigloyloxytropane (Fig. 1). We also analyzed the tissue expression pattern of *TS*, observed its subcellular localization, studied its metabolic roles *in planta*, and analyzed its catalytic activities and mechanism. Finally, 3β-tigloyloxytropane was efficiently produced in engineered *E. coli*. This identification of TS enriches the knowledge on the biosynthesis of TAs and provides a biotechnological approach to produce 3β-tropanol-derived alkaloids through synthetic biology.

## Results

### The expression of *TS* is associated with the 3β-tigloyloxytropane distribution in *A. belladonna*

To identify the genes responsible for 3β-tigloyloxytropane biosynthesis, we first analyzed the levels of 3β-tigloyloxytropane in different

**Fig. 1 | Proposed pathway for the biosynthesis of tropanol-derived alkaloids.** TRI tropine-forming reductase or tropinone reductase I, TRII pseudotropine-forming reductase or tropinone reductase II, LS littorine synthase, TS tigloidine synthase, the key finding of this study. The 3β-tropane alkaloid biosynthetic pathway is marked in blue, and the 3α-tropane alkaloid biosynthetic pathway is marked in green. P450-5021, 3β-tigloyloxytropane demethylase. P450-116623, tigloyl norpseudotropine hydroxylase.

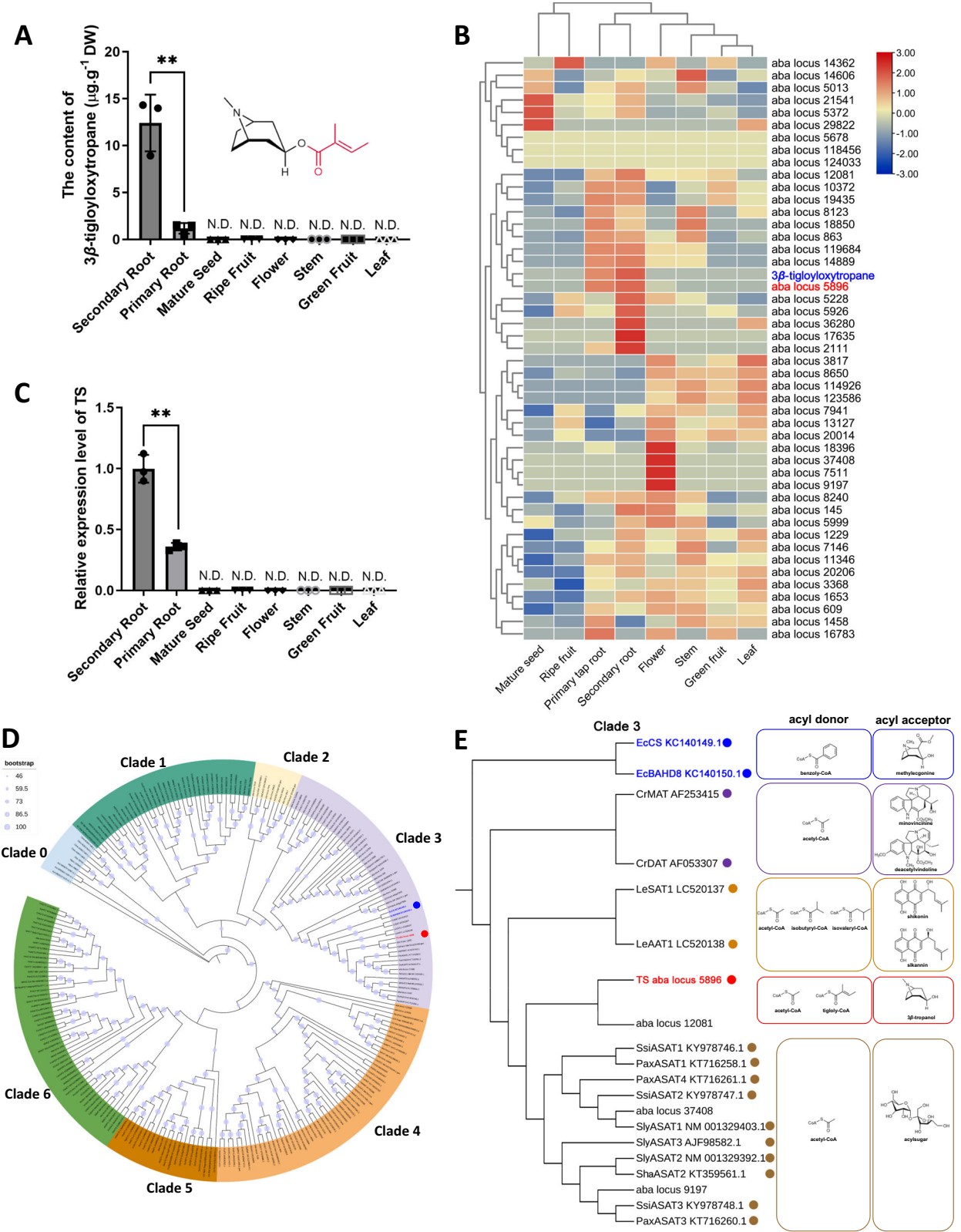

organs of *A. belladonna*. 3β-Tigloyloxytropane was detected in the primary and secondary roots, but not in any other tissues of *A. belladonna*, including mature seeds, ripe fruits, flowers, stems, green fruits, and leaves (Fig. 2A). With respect to 3β-tigloyloxytropane, 1.18 μg.g⁻¹ dry weight (DW) and 12.40 μg.g⁻¹ DW were detected in primary and secondary roots, respectively. The root-specific distribution of 3β-tigloyloxytropane suggested that the corresponding genes exhibited

similar expression patterns, i.e., root-specific or root-preferring. Through HMMER search analysis, 46 BAHD-AT genes were identified from the *A. belladonna* transcriptome. Subsequently, metabolite and gene expression association analysis (MGAA) was used to screen out candidates that produce 3β-tigloyloxytropane. The MGAA results revealed that a putative BAHD gene (aba locus 5896), named *TS*, was specifically expressed in primary and secondary roots and thus was

**Fig. 2 | Metabolite, transcriptome, and phylogenetic association analysis.**
**A** Tissue profile of 3$\beta$-tigloyloxytropane. Three independent plants were used in the tissue profile analysis of 3$\beta$-tigloyloxytropane. The data are presented as means values ± s.d. **P = 0.0032. DW, dry weight. **B** Association analysis of the metabolites and BAHD gene family expression patterns. **C** Relative expression levels of TS in different organs, as indicated by qPCR. Three independent plants were used in the relative expression analysis of TS. The data are presented as means values ± s.d. **P = 0.0007. **D** Phylogenetic analysis of the BAHD-AT gene family. The phylogenetic tree was divided into clades 0–6. **E** BAHD acyltransferases and their substrates in clade 3: TS, the BAHD acyltransferase identified in this study; EcCS, *Erythroxylum coca* cocaine synthase, a BAHD acyltransferase of the coca tree; EcBAHD8, the

homolog of EcCS in *Erythroxylum coca*; PaxASAT1-4, an acylsugar acyltransferase in *Petunia axillaris*; SlyASAT1-3, an acylsugar acyltransferase in *Solanum lycopersicum*; SsiASAT1-3, an acylsugar acyltransferase in *Salpiglossis sinuata*; ShaASAT2, an acylsugar acyltransferase in *Solanum habrochaites*; LeSAT1, a shikonin *O*-acyltransferase in *Lithospermum erythrorhizon*; LeAAT1, an alkannin *O*-acyltransferase in *Lithospermum erythrorhizon*; CrMAT, a minovincinine-19-hydroxy-*O*-acetyltransferase in *Catharanthus roseus*; and CrDAT, a deacetylvindoline 4-*O*-acetyltransferase in *Catharanthus roseus*. All functionally identified BAHD-ATs are labeled with their corresponding accession numbers. Statistical analysis was performed according to the two-sided independent sample *t*-test.

tightly clustered with 3$\beta$-tigloyloxytropane (Fig. 2B). The expression pattern of *TS* was confirmed by qPCR after the full-length cDNA sequence was isolated (Fig. 2C).

## Phylogenetic analysis of BAHD acyltransferases

To determine the evolutionary relationship of TS, a phylogenetic tree of the BAHD-AT family was constructed, which included 46 BAHD-ATs of *A. belladonna* and 196 functionally identified BAHD-ATs from other plants (Fig. 2D and Supplementary Data 1–2). The phylogenetic tree was divided into clades 0–6, and these clades were the same as those in a previous report[35]. Phylogenetic analysis revealed that TS were clustered into clade 3 (Fig. 2D). In clade 3, TS and functionally identified BAHD-ATs, such as PaASAT1-4, SlyASAT1-3, LeSAT1, LeAAT1, CaMAT and CaDAT, were included (Fig. 2E). These BAHDs recognize short-chain acyl-CoA thioesters, including acetyl-CoA, isobutyryl CoA, and isovaleryl-CoA, as acyl donors[36–39]. Nonetheless, EcCS and EcBAHD8 were also included in clade 3 (Fig. 2E); these two BAHD enzymes catalyze cocaine formation using methylecgonine with a 3$\beta$-tropane ring skeleton as the acyl acceptor[32]. Therefore, these results suggest that TS recognizes acyl acceptors with a 3$\beta$-tropane ring skeleton, and acyl donors of short-chain acyl-CoA thioesters.

## TS catalyzes the esterification of 3$\beta$-tropanol

To investigate the function of TS, we produced recombinant TS proteins in engineered *E. coli* and purified. First, we synthesized 3$\beta$-tropanol esters and tigloyl-CoA using chemical methods (Supplementary Figs. S1–S24 and Supplementary Data 3). In a reaction system with 3$\beta$-tropanol and tigloyl-CoA, TS produced a product with an $[M + H]^+$ m/z of 224.1645, identical to that of the authentic 3$\beta$-tigloyloxytropane (Fig. 3A, B). Nonetheless, when TS was boiled, no products were detected (Fig. 3A). In a reaction system with 3$\alpha$-tropanol and tigloyl-CoA, TS did not generate products (Fig. 3A). The above results indicated that TS catalyzes the condensation of 3$\beta$-tropanol with tigloyl-CoA to generate 3$\beta$-tigloyloxytropane via esterification reactions.

Furthermore, the optimum pH and temperature conditions for TS catalysis were investigated. The catalytic activities of TS treated with 3$\beta$-tropanol and tigloyl-CoA in the pH range of 6.0–10.6. The maximum activity of TS was detected at pH 8.6 (Supplementary Fig. S25A). The optimal pH conditions for the catalytic activity of most BAHD proteins are in a narrow alkaline range, approximately from 8 to 10. For example, the optimal pH conditions for EcCS, CiHCT2, and NtMAT1 are 9.4, 9.0, and 8.5, respectively[32,40,41]. In addition, the catalytic activities of TS treated with 3$\beta$-tropanol and tigloyl-CoA between 20 °C and 50 °C were determined (Supplementary Fig. S25B). The maximum activity of TS was detected at 30 °C. Next, the enzyme kinetic constants were measured under the optimal conditions (pH 8.6 and 30 °C).

BAHD-AT family proteins generally exhibit substrate promiscuity[32,42–45]. Therefore, we tested different acyl donors for TS. When 3$\beta$-tropanol and acetyl-CoA were used as substrates, TS produced 3$\beta$-acetoxytropane with a $[M + H]^+$ m/z value of 184.1332 (Supplementary Fig. S26). When 3$\beta$-tropanol and benzoyl-CoA were used as substrates, TS produced 3$\beta$-benzoyloxytropane with a $[M + H]^+$ m/z value of 246.1489 (Supplementary Fig. S27) in trace amounts. When

tigloyl-CoA, acetyl-CoA, and benzoyl-CoA were used as acyl donors, the $K_m$ values of TS for 3$\beta$-tropanol were 0.36 mM, 0.39 mM, and 0.43 mM, respectively (Table 1). Moreover, no significant difference was observed between these results. However, the $K_m$ values of TS for tigloyl-CoA, acetyl-CoA, and benzoyl-CoA were significantly different at 0.02 mM, 0.09 mM, and 0.92 mM respectively (Table 1). These results suggested that TS has a greater affinity for tigloyl-CoA than for acetyl-CoA or benzoyl-CoA. The catalytic efficiency ($K_{cat}/K_m$) of TS for accessing tigloyl-CoA was 338332.70 $M^{-1}.S^{-1}$, which was 743 and 26473 times greater than the amounts of acetyl-CoA and benzoyl-CoA, respectively (Table 1). The enzymatic assays indicated that TS mainly catalyzes the formation of 3$\beta$-tigloyloxytropane.

## Silencing of *TS* disrupts the biosynthesis of calystegine A3 *in planta*

To determine the role of *TS* in the biosynthesis of 3$\beta$-tropanol esters *in planta*, *TS* was suppressed by virus-induced gene silencing (VIGS) in *A. belladonna* seedlings. As shown in Fig. 3C, *TS* was effectively suppressed in *A. belladonna* roots. In accordance with the decreased expression level of TS, the contents of 3$\beta$-tropanol, 3$\alpha$-tropanol, and tropanol hexosides in the TS-silenced lines were markedly greater than those in the control lines (Fig. 3D and Supplementary Fig. S28). The contents of 3$\beta$-tigloyloxytropane, 3$\beta$-acetoxytropane, tigloyl norpseudotropine, tigloyl 1-hydroxynorpseudotropine, and calystegine A3 in the TS-silenced lines were markedly lower than those in the control lines (Fig. 3E–I). However, the contents of the 3$\alpha$-tropanol derivatives littorine, hyoscyamine, and scopolamine, in *TS*-silenced plants did not significantly differ from those in the control plants (Supplementary Fig. S28). Both in-planta and in vitro assays showed that TS catalyzes the formation of 3$\beta$-tigloyloxytropane in the biosynthetic pathway of calystegine A3.

## Overexpression of *TS* increases the content of 3$\beta$-tigloyloxytropane in hairy roots

In addition, we overexpressed *TS* in hairy root cultures of *A. belladonna* to further investigate its role in the biosynthesis of 3$\beta$-tigloyloxytropane and explore its application in engineering. Genomic-PCR and qPCR analysis indicated that *TS* was integrated into the genome of *A. belladonna* and that its expression level was greatly increased in *TS*-over-expressing root lines (Fig. 3J, Supplementary Fig. S29). The contents of 3$\beta$-tigloyloxytropane and 3$\beta$-acetoxytropane were significantly greater in the *TS*-overexpressing lines than in the control lines (Fig. 3L, M), while no differences were observed in the production of 3$\beta$-tropanol, tigloyl norpseudotropine, tigloyl 1-hydroxynorpseudotropine and calystegine A3 (Fig. 3K, N–P). In addition, the contents of 3$\alpha$-tropanol derivatives in the *TS*-overexpressing lines were not significantly different from those in the control lines (Supplementary Fig. S30). Overall, these results indicated that TS overexpression could increase the production of 3$\beta$-tigloyloxytropane in *A. belladonna* hairy roots.

## TS is localized in the mitochondria

Previous studies reported that BAHD-ATs are localized in the cytosol[29,31,46]. When analyzing the targeting signal sequence in the TS,

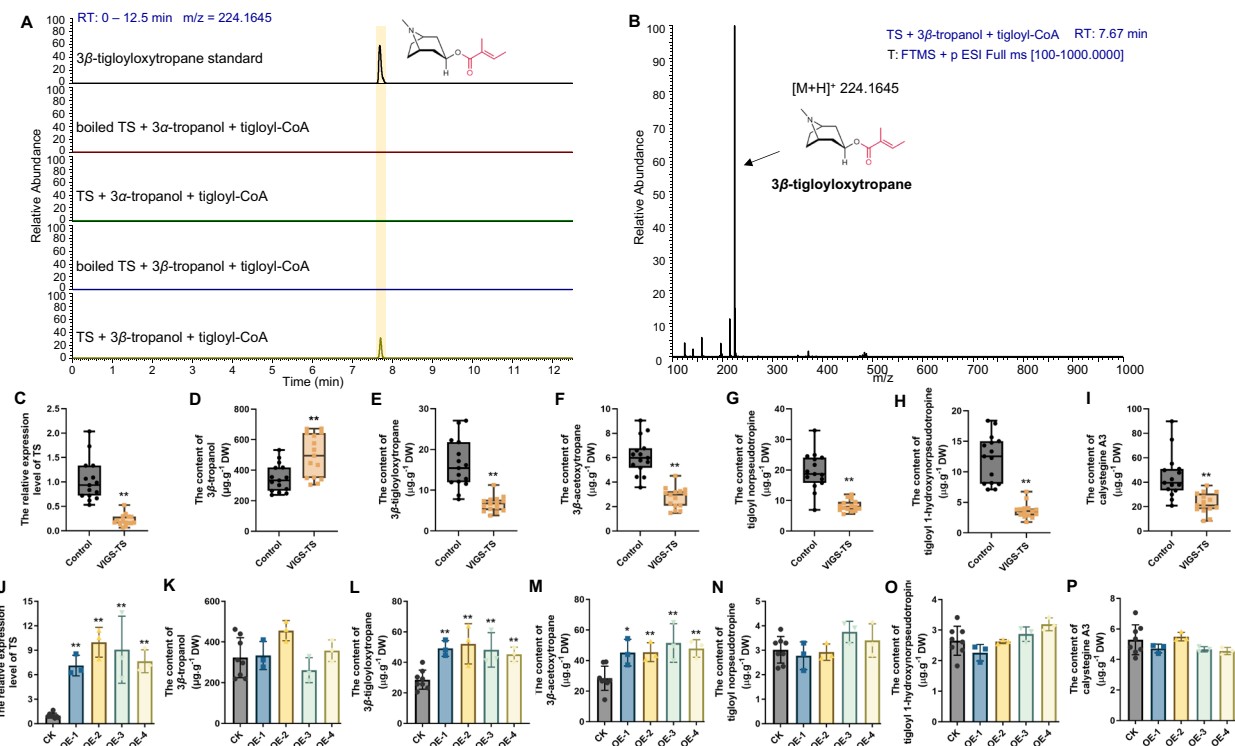

**Fig. 3 | Functional characterization of TS. A** TS enzymatic assays with tigloyl-CoA as the acyl donor and 3$\beta$-tropanol as the acyl acceptor. **B** Mass spectrometry (MS) data of 3$\beta$-tigloyloxytropane. **C** Relative expression levels of *TS* in VIGS-*TS A. belladonna* seedlings. **P < 0.0001. **D** Contents of 3$\beta$-tropanol in VIGS-*TS A. belladonna* seedlings. **P = 0.0024. **E** Contents of 3$\beta$-tigloyloxytropane in VIGS-*TS A. belladonna* seedlings. **P < 0.0001. **F** Contents of 3$\beta$-acetoxytropane in VIGS-*TS A. belladonna* seedlings. **P < 0.0001. **G** Contents of tigloyl norpseudotropine in VIGS-*TS A. belladonna* seedlings. **P < 0.0001. **H** Contents of tigloyl 1-hydroxynorpseudotropine in VIGS-TS *A. belladonna* seedlings. **P < 0.0001. **I** Contents of calystegine A3 in VIGS-TS *A. belladonna* seedlings. **P = 0.0002. Control, control line obtained by empty plasmid transformation. VIGS-*TS*, TS-silenced line. Fifteen independent plants each from the control group and VIGS-*TS* group were used in the VIGS assays. Center line of box plot denotes the median value; lower and upper bounds denote first and third quartile; whiskers extend to the smallest and maximum values. **J** Relative expression levels of *TS* in *TS*-overexpressing *A. belladonna* hairy root cultures. **P < 0.0001 (OE−1), **P < 0.0001 (OE-2), **P < 0.0001 (OE-3), **P < 0.0001 (OE-4). **K** Contents of 3$\beta$-tropanol in *TS*-overexpressing *A. belladonna* hairy root cultures. **L** Contents of 3$\beta$-tigloyloxytropane in *TS*-overexpressing *A. belladonna* hairy root cultures. **P = 0.0007 (OE−1), **P = 0.0024 (OE-2), **P = 0.0044 (OE-3), **P = 0.0025 (OE-4). **M** Contents of 3$\beta$-acetoxytropane in *TS*-overexpressing *A. belladonna* hairy root cultures. *P = 0.0136 (OE-1), **P = 0.0087 (OE-2), **P = 0.0048 (OE-3), **P = 0.0039 (OE-4). **N** Contents of tigloyl norpseudotropine in *TS*-overexpressing *A. belladonna* hairy root cultures. **O** Contents of tigloyl 1-hydroxynorpseudotropin in *TS*-overexpressing *A. belladonna* hairy root cultures. **P** Contents of calystegine A3 in *TS*-overexpressing *A. belladonna* hairy root cultures. CK, eight independently transformed root culture lines transformed with pBI121. OE denotes all independently transformed root culture lines overexpressing TS (three biological replicates for each line), including OE-1, OE-2, OE-3, and OE-4. The data are presented as means values ± s.d. Statistical analysis was performed according to the two-sided independent sample *t*-test. DW, dry weight.

---

we determined that the TS contains a potential mitochondrion-localizing signal sequence in its *N*-terminus (Supplementary Fig. S31). The subcellular localization of TS was subsequently investigated in tobacco protoplasts. TS fused with YFP (TS-YFP) and its mitochondrion-localizing signal peptide (the 32 amino acids at the N-terminus) fused with YFP (N32-YFP) were expressed in tobacco protoplasts. YFP signals in tobacco protoplasts overlapped with those derived from the mitochondrion-specific fluorescent dye Mito-Tracker Red (Fig. 4A). Furthermore, we performed a subcellular localization analysis of TS without the 32 amino acids at its N-terminus. Based on the results, TS without 32 amino acids (TS^Del-N32) still exhibited mitochondrial localization (Supplementary Fig. S32). Therefore, we speculate that the mitochondrial localization signal of TS is not only located at its N-terminus but also distributed at other regions[47–51]. When TS^Del-N32 was expressed in *E. coli*, its catalytic activity decreased markedly (Supplementary Fig. S33A), suggesting that the 32-amino-acid sequence of TS plays a crucial role in enzymatic activity. Based on the 3D structure of TS generated by Alpha-Fold2, its 32-amino-acid sequence constitutes the core scaffold of TS (Supplementary Fig. S33B).

Next, we isolated mitochondria and supernatant from the secondary roots of *A. belladonna*. Voltage-dependent anion channel (VDAC) was used to detect mitochondria[52–54]. The mitochondrion marker protein VDAC was detected in the mitochondrial fraction but not in the supernatant fraction (Fig. 4B). These results suggested that we successfully obtained mitochondria from *A. belladonna*. The catalytic activities of the crude protein from the mitochondria and supernatant were subsequently detected. The formation of 3$\beta$-tigloyloxytropane was catalyzed by the crude mitochondrial protein catalyzed but not the crude protein in the supernatant (Fig. 4C).

These results demonstrated that TS are localized mitochondria; this subcellular compartmentalization is different from that of previously reported BAHD-ATs, which are localized in the cytosol[29].

### Catalytic mechanism of TS and improvements of TS inactivity
The catalytic mechanisms of BAHD-ATs with phenylpropanoids as acyl acceptors have been studied extensively[55–58]; however, the mechanism through which BAHD-ATs recognize 3$\beta$-tropanol, a compound with very different structures from phenylpropanoids, as an acyl acceptor,

**Table 1 | Kinetic parameters of TS**

| Product | Substrate | Enzyme | $K_m$ (mM) | $K_{cat}$ (s$^{-1}$) | $K_{cat}/K_m$ (M$^{-1}$·s$^{-1}$) |
|---|---|---|---|---|---|
| 3$\beta$-tigloyloxytropane | 3$\beta$-tropanol | TS | 0.36 ± 0.05 | 6.84 ± 0.30 | 19045.07 ± 1658.79 |
| 3$\beta$-tigloyloxytropane | 3$\beta$-tropanol | TS$^{S40T}$ | 0.32 ± 0.07 | 11.09 ± 0.81 | 35153.35 ± 5480.42 |
| 3$\beta$-tigloyloxytropane | 3$\beta$-tropanol | TS$^{F46I}$ | 0.32 ± 0.06 | 1.44 ± 0.09 | 4550.90 ± 593.97 |
| 3$\beta$-tigloyloxytropane | 3$\beta$-tropanol | TS$^{S40T-F46I}$ | 0.27 ± 0.07 | 2.48 ± 0.22 | 9464.65 ± 1784.82 |
| 3$\beta$-tigloyloxytropane | tigloyl-CoA | TS | 0.02 ± 0.004 | 7.38 ± 0.25 | 338332.70 ± 49061.33 |
| 3$\beta$-tigloyloxytropane | tigloyl-CoA | TS$^{S40T}$ | 0.02 ± 0.003 | 11.42 ± 0.34 | 572303.6 ± 72285.97 |
| 3$\beta$-tigloyloxytropane | tigloyl-CoA | TS$^{F46I}$ | 0.41 ± 0.11 | 1.42 ± 0.17 | 3588.34 ± 569.56 |
| 3$\beta$-tigloyloxytropane | tigloyl-CoA | TS$^{S40T-F46I}$ | 0.15 ± 0.02 | 2.27 ± 0.10 | 14809.27 ± 1370.76 |
| 3$\beta$-acetoxytropane | 3$\beta$-tropanol | TS | 0.39 ± 0.11 | 0.048 ± 0.005 | 126.03 ± 23.98 |
| 3$\beta$-acetoxytropane | 3$\beta$-tropanol | TS$^{S40T}$ | 0.31 ± 0.08 | 0.098 ± 0.007 | 319.69 ± 52.89 |
| 3$\beta$-acetoxytropane | 3$\beta$-tropanol | TS$^{F46I}$ | 0.48 ± 0.08 | 0.0008 ± 0.0004 | 16.79 ± 1.64 |
| 3$\beta$-acetoxytropane | 3$\beta$-tropanol | TS$^{S40T-F46I}$ | 0.53 ± 0.08 | 0.02 ± 0.001 | 41.03 ± 3.67 |
| 3$\beta$-acetoxytropane | acetyl-CoA | TS | 0.09 ± 0.02 | 0.04 ± 0.002 | 455.09 ± 62.68 |
| 3$\beta$-acetoxytropane | acetyl-CoA | TS$^{S40T}$ | 0.07 ± 0.01 | 0.09 ± 0.004 | 1191.44 ± 155.40 |
| 3$\beta$-acetoxytropane | acetyl-CoA | TS$^{F46I}$ | 0.35 ± 0.12 | 0.005 ± 0.0008 | 16.17 ± 3.42 |
| 3$\beta$-acetoxytropane | acetyl-CoA | TS$^{S40T-F46I}$ | 0.27 ± 0.03 | 0.02 ± 0.0009 | 73.93 ± 5.33 |
| 3$\beta$-benzoyloxytropane | 3$\beta$-tropanol | TS | 0.43 ± 0.05 | 0.006 ± 0.0003 | 14.55 ± 1.18 |
| 3$\beta$-benzoyloxytropane | 3$\beta$-tropanol | TS$^{S40T}$ | 0.46 ± 0.09 | 0.02 ± 0.001 | 37.09 ± 4.52 |
| 3$\beta$-benzoyloxytropane | 3$\beta$-tropanol | TS$^{F46I}$ | 0.34 ± 0.06 | 0.03 ± 0.002 | 90.78 ± 10.21 |
| 3$\beta$-benzoyloxytropane | 3$\beta$-tropanol | TS$^{S40T-F46I}$ | 0.48 ± 0.09 | 0.07 ± 0.005 | 155.33 ± 17.65 |
| 3$\beta$-benzoyloxytropane | benzoyl-CoA | TS | 0.92 ± 0.58 | 0.01 ± 0.004 | 12.78 ± 3.57 |
| 3$\beta$-benzoyloxytropane | benzoyl-CoA | TS$^{S40T}$ | 0.84 ± 0.38 | 0.03 ± 0.006 | 32.73 ± 6.77 |
| 3$\beta$-benzoyloxytropane | benzoyl-CoA | TS$^{F46I}$ | 0.37 ± 0.09 | 0.04 ± 0.004 | 99.32 ± 13.33 |
| 3$\beta$-benzoyloxytropane | benzoyl-CoA | TS$^{S40T-F46I}$ | 0.26 ± 0.08 | 0.08 ± 0.008 | 304.94 ± 54.07 |

has remained unknown. To explore the catalytic mechanism of TS and provide information for engineering, a complex model was constructed by molecular docking (Supplementary Fig. S34). We observed that His162 in the HXXXD motif forms a hydrogen bond with the oxygen of 3$\beta$-tropanol (3.0 Å), which is a universally conserved catalytic residue in BAHD-ATs, suggesting that His162 is the base catalyst in the catalytic pocket of TS (Fig. 5A). Consistent with our hypothesis, when His162 was mutated to alanine, TS$^{H162A}$ completely lost its catalytic activity (Fig. 5B). In addition, a binding pocket of 3$\beta$-tropanol formed by His162, Ile35, Gln39, Asn298, Leu300, Tyr280 and Trp340 was observed (Fig. 5C). To examine the roles of these residues in the binding pocket, site-directed mutants of TS were generated. Ile35, Gln39, Tyr280, Asn298, Leu300, and Trp340 were mutated to alanine, and the catalytic activities of the mutants were dramatically reduced (Fig. 5D). Together, our findings revealed that these residues in the binding pocket of 3$\beta$-tropanol play critical roles in governing TS activity.

The results of enzymatic kinetic analysis indicated that TS poorly utilizes benzoyl-CoA, suggesting that TS prefers short-chain acyl-CoA thioesters as acyl donors rather than benzoyl-CoA with an aromatic amino acid. The mechanism by which TS strictly govern acyl donor identification is interesting. We assumed that the greater steric hindrance of benzoyl-CoA relative to short-chain acyl-CoA thioesters, which is not suitable for the substrate pocket of TS, contributed to the low production of 3$\beta$-benzoyloxytropane. Through molecular docking, we compared the substrate pocket of TS with that of EcCS, which uses benzoyl-CoA as an acyl donor. A Phe46 residue was observed in TS that is 2.5 Å away from the benzene ring of benzoyl-CoA (Fig. 6A), whereas its corresponding residue in EcCS (Ile45) is 4.5 Å (Fig. 6C). When Phe46 in TS was mutated to isoleucine, the distance between the benzene ring of benzoyl-CoA and Ile46 was 4.1 Å (Fig. 6B). When the 46th amino acid of TS is phenylalanine, the distances to tigloyl-CoA and acetyl-CoA are 3.6 Å and 5.0 Å, respectively (Fig. 6D, E). However,

when the 46th amino acid of TS is isoleucine, the distances increase to 4.9 Å and 6.3 Å, respectively (Fig. 6F, G). Subsequently, we conducted in vitro enzyme assays to verify the results of the molecular docking experiments. Consistent with our hypothesis, the F46I mutation significantly enhanced the activity of TS in the synthesis of 3$\beta$-benzoyloxytropane (Fig. 6H and Table 1) and significantly decreased the activity of TS in the synthesis of 3$\beta$-tigloyloxytropane (Table 1). The $K_{cat}/K_m$ value for benzoyl-CoA in TS$^{F46I}$ was 99.32 M$^{-1}$·s$^{-1}$, which was 7.77 times greater than that of TS (Table 1). The $K_{cat}/K_m$ values for tigloyl-CoA and acetyl-CoA in TS$^{F46I}$ were 3588.34 M$^{-1}$·s$^{-1}$ and 16.17 M$^{-1}$·s$^{-1}$, which were 1.06% and 3.5%, respectively, of the TS (Table 1). The results above showed that Phe46 is the key amino acid residue that governs the use of short-chain acyl-CoA thioesters as acyl donors.

To further improve the catalytic activity of TS, a consensus protein design was employed. Based on the conservative substitution analysis performed on residues within a 5 Å range of His162, we observed that only Ser40 did not fit the conservative replacement model TS (Supplementary Fig. S35A). Virtual mutation prediction revealed that substituting Ser40 with the conservative substitution threonine may change the hydrogen bonding network around the 3$\beta$-tropanol entry channel (Supplementary Fig. S35B, C). Thr40 established a new hydrogen bond with Asn135 in the adjacent coil (Supplementary Fig. S35B, C). Similarly, the $K_{cat}$ values of TS$^{S40T}$ for synthesizing 3$\beta$-tropanol esters were significantly greater than those of the other samples, while the $K_m$ values were not significantly different (Table 1). The $K_{cat}/K_m$ values for 3$\beta$-tropanol and tigloyl-CoA in TS$^{S40T}$ were 35153.35 M$^{-1}$·s$^{-1}$ and 572303.6 M$^{-1}$·s$^{-1}$, which was 3.00 and 1.69 times greater than that of the wild-type TS (Table 1 and Supplementary Fig. S35D, E). The results above demonstrated that compared to wild-type TS, the TS$^{S40T}$ mutant could synthesize 3$\beta$-tropanol esters, including 3$\beta$-tigloyloxytropane, 3$\beta$-acetoxytropane and 3$\beta$-benzoyloxytropane, more efficiently.

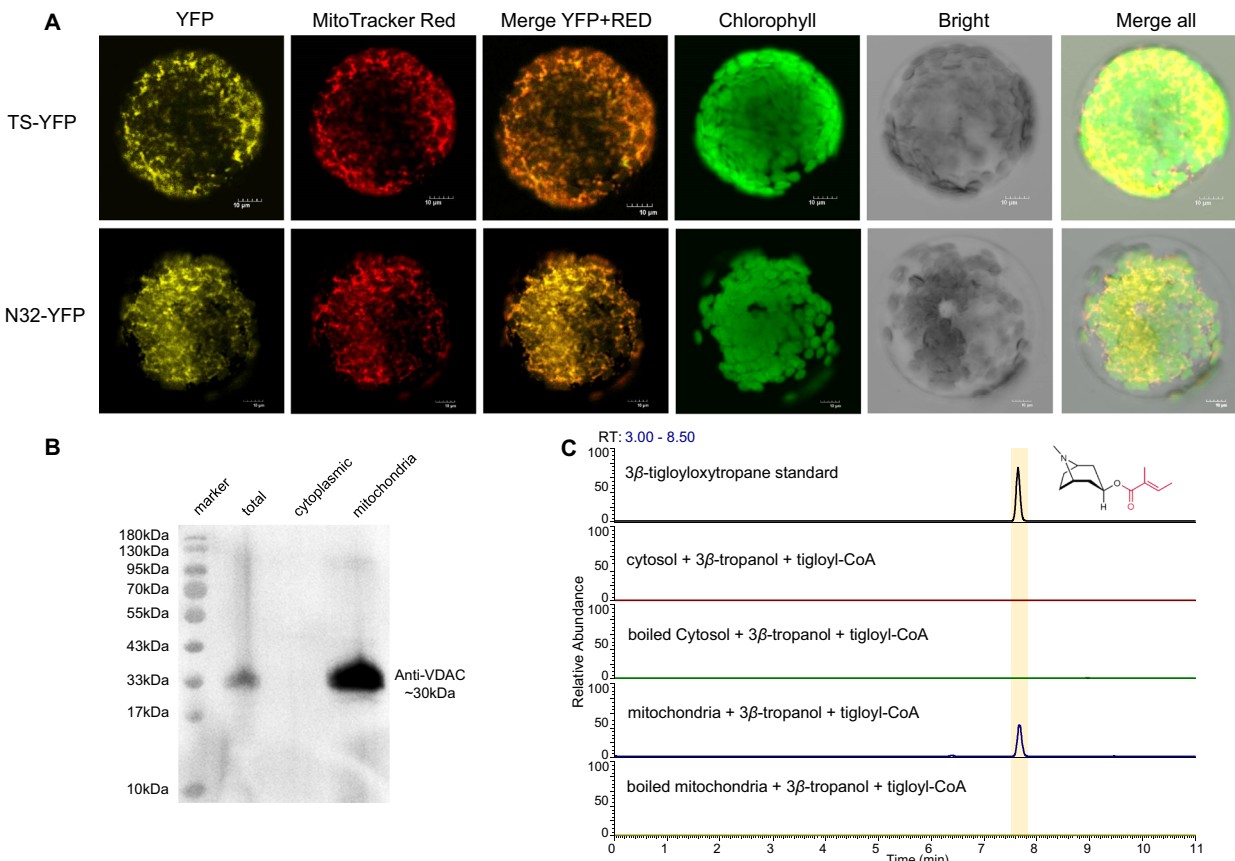

**Fig. 4 | Subcellular localization analysis of TS. A** The localization of TS-YFP and N32-YFP in tobacco protoplasts was observed by confocal microscopy. YFP yellow fluorescence from YFP. MitoTracker Red, MitoTracker Red fluorescence-labeled mitochondria. Merge YFP + RED, the merged images for the yellow fluorescence and MitoTracker Red fluorescence. Chlorophyll, chlorophyll spontaneous fluorescence. Bright, bright field image. Overlapping images of all the channels mentioned above were merged. TS-YFP, TS fused with YFP. N32-YFP, mitochondrion-localizing signal peptide (the 32 amino acids at the N-terminus) of TS fused with YFP. All tobacco transformation and microscopic analyses were independently conducted three times with different plants. **B** Determination of *A. belladonna* mitochondria by Western blot analysis with an antibody against VDAC. Western blot analysis was independently conducted two times with different plants. **C** Crude protein from *A. belladonna* mitochondria-catalyzed tigloyl-CoA and 3β-tropanol to form 3β-tigloyloxytropane.

## Biosynthesis of 3β-tigloyloxytropane in *N. benthamiana*

To further assess the function of TS and its application in 3β-tigloyloxytropane engineering in tobacco, we reconstructed the 3β-tropanol ester biosynthetic pathway in the leaves of *N. benthamiana*. Six upstream biosynthesis genes (*Erythroxylum novogranatense* ornithine decarboxylase, EnODC; *A. belladonna* putrescine *N*-methyltransferase, AbPMT; *A. belladonna* *N*-methylputrescine oxidase, AbMPO; *A. belladonna* type III polyketide synthase, AbPYKS; *A. belladonna* tropinone synthase, AbCYP82M3; and *Datura stramonium* tropinone reductase II, DsTRII), together with TS were transiently coexpressed in tobacco leaves (Fig. 7A). Control experiments were conducted in which yellow fluorescent protein (YFP) was expressed. Leaves that expressed these six enzymes and TS yielded tropinone, hygrine, 3β-tropanol, 3β-tigloyloxytropane and 3β-acetoxytropane (Fig. 7B–G). The control leaves did not yield any 3β-tigloyloxytropane or intermediates. To further promote the harvest of 3β-tigloyloxytropane from tobacco leaves, we substituted the wild-type TS with $TS^{S40T}$. $TS^{S40T}$ increased the levels of 3β-tigloyloxytropane and 3β-acetoxytropane by 1.33- and 5.02- fold respectively, compared with those in the wild-type TS. The highest contents of 3β-tigloyloxytropane and 3β-acetoxytropane were detected in tobacco leaves, which were 6.14 µg.g⁻¹ DW and 0.13 µg.g⁻¹ DW, respectively (Fig. 7F, G).

Although tobacco leaves coexpressed the seven biosynthetic genes involved in 3β-tigloyloxytropane biosynthesis, the production level was low. This low yield might result from the insufficient supply of tigloyl-CoA in plants. Increasing the synthesis of tigloyl-CoA may promote the accumulation of 3β-tigloyloxytropane. Tigloyl-CoA is a metabolite degraded from isoleucine in plants[59,60], but metabolic genes related to tigloyl-CoA formation have not been identified.

We hypothesized that CoA ligase catalyzes the formation of tigloyl-CoA using tiglic acid and CoA. In bacteria, *Pseudomonas chlororaphis* contains an isobutyryl CoA synthetase (PcICS) that catalyzes a reaction between isobutyric acid and CoA to produce isobutyryl CoA[61]. We hypothesized that PcICS uses tiglic acid as a substrate because the structures of isobutyric acid and tiglic acid are similar. Consistent with this hypothesis, purified PcICS catalyzed the synthesis of tigloyl-CoA from CoA and tiglic acid (Fig. 8A, B). Unfortunately, when the six upstream biosynthesis genes, $TS^{S40T}$ and PcICS were coexpressed and tiglic acid was added to tobacco leaves, the level of 3β-tigloyloxytropane did not significantly increase (Supplementary Fig. S36), probably due to the insufficient supply of 3β-tropanol.

Then, we performed further experiments. When TS was expressed in tobacco leaves fed 3β-tropanol, 3β-tigloyloxytropane was produced at a much greater level than that produced during the de novo biosynthesis of 3β-tigloyloxytropane (Supplementary Fig. S37). When TS was expressed in tobacco leaves fed sufficient 3β-tropanol and tiglic acid, 3β-tigloyloxytropane was produced at a slightly increased level (Supplementary Fig. S37). When PcICS and $TS^{S40T}$ were coexpressed in tobacco leaves fed 3β-tropanol and tiglic acid, 3β-tigloyloxytropane was produced at markedly increased levels, reaching up to

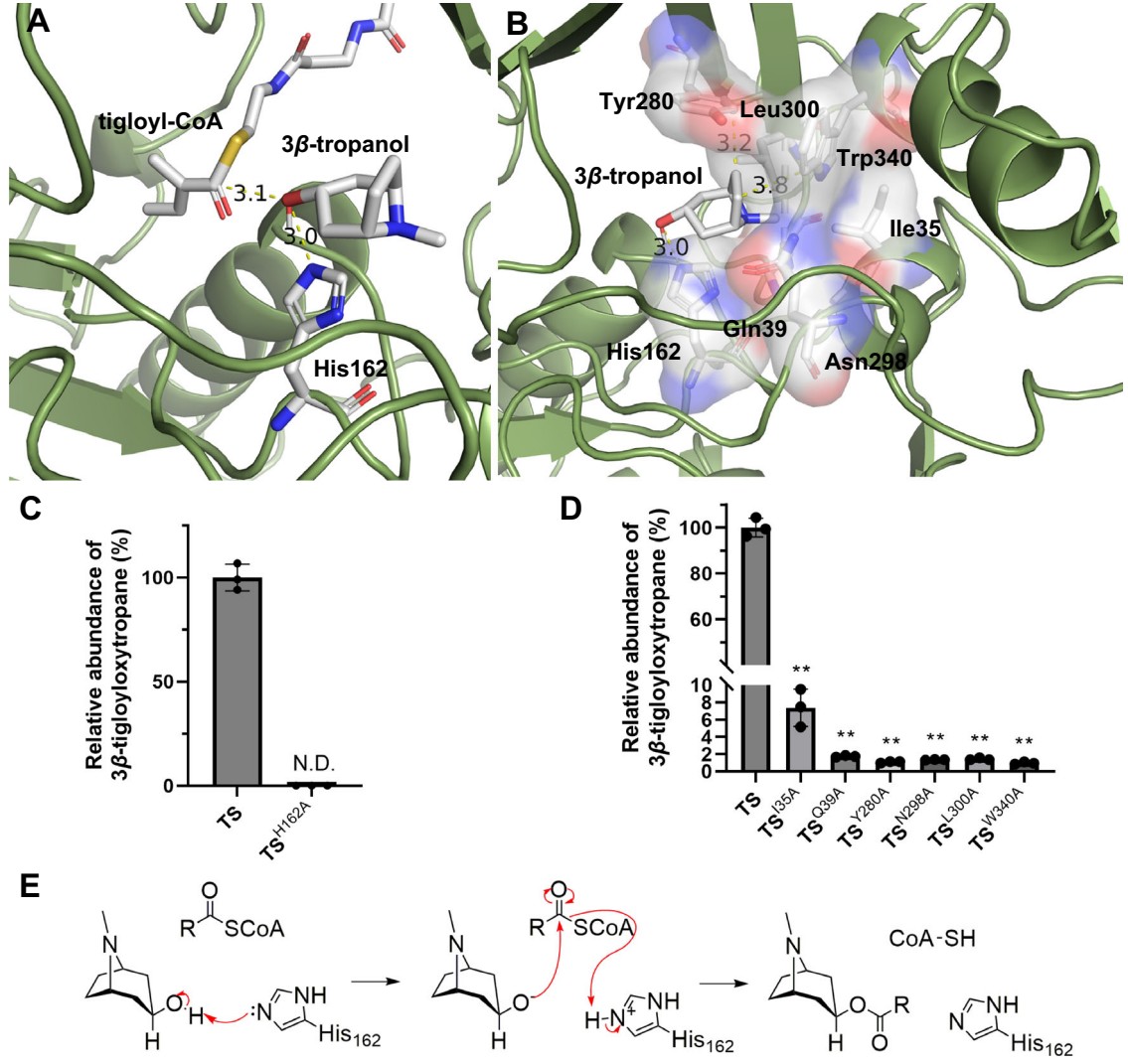

**Fig. 5 | Catalytic mechanism of TS. A** Schematic model of the catalytic pocket that contains 3β-tropanol and tigloyl-CoA. **B** Comparison of the relative activities of TS and TS$^{H162A}$ using tigloyl-CoA as the acyl donor. **C** Key residues in the substrate pocket combined with 3β-tropanol. **D** Comparison of the relative activity of TS and TS mutants (3β-tropanol binding residues mutated to alanine) using tigloyl-CoA as the acyl donor. **P < 0.0001 (TS$^{I35A}$), **P < 0.0001 (TS$^{Q39A}$), **P < 0.0001 (TS$^{Y280A}$),

**P < 0.0001 (TS$^{N298A}$), **P < 0.0001 (TS$^{L300A}$), **P < 0.0001 (TS$^{W340A}$). **E** 3β-Tropanol and acyl donors are converted to 3β-tropanol esters via the catalysis of TS. The data are presented as means values ± s.d. Recombinant protein obtained from three independent transformants of TS and each mutant for activity test. Statistical analysis was performed according to the two-sided independent sample $t$-test.

293.86 μg.g$^{−1}$ DW (Supplementary Fig. S37). These results suggested that a sufficient supply of substrates through feeding facilitated the production of 3β-tigloyloxytropane in tobacco leaves.

**Biosynthesis of 3β-tigloyloxytropane in *E. coli***

Due to the low production of 3β-tigloyloxytropane in plants, we constructed an efficient platform to produce this potential drug for treating neurodegenerative diseases. We therefore built an engineered *E. coli* bioreactor to produce 3β-tigloyloxytropane, by feeding the readily available substrates tiglic acid and 3β-tropanol (Fig. 8C).

To produce 3β-tigloyloxytropane, *E. coli* cells expressing TS were fermented in LB medium with 250 mg.L$^{−1}$ of 3β-tropanol. Although TS and TS$^{S40T}$ catalyzed the production of 3β-tropanol and tigloyl-CoA to 3β-tigloyloxytropane in plants and in vitro, we did not detect 3β-tigloyloxytropane in this fermentation system (Fig. 8D). This difference might be attributed to the absence of tigloyl-CoA in *E. coli*. Therefore, to produce 3β-tigloyloxytropane, a CoA ligase that catalyzes the synthesis of tigloyl-CoA from CoA and tiglic acid is necessary. Thus, PcICS was used to engineer *E. coli* to produce 3β-

tigloyloxytropane together with TS or its mutants. The highest yield (357.40 mg.L$^{−1}$ with a conversion rate of 90.9%), was detected after 60 h of fermentation when TS$^{S40T}$ and PcICS were expressed (Fig. 8D).

## Discussion

Calystegines are potential anti-nutritional factors that have been found in various Solanaceous foods, such as potatoes and tomatoes[23]. Recently, the discovery of two P450s involved in the biosynthesis of calystegines shed the light on the complete elucidation of the biosynthetic pathway of calystegines[26]. However, the crucial esterification step leading to the formation of 3β-tigloyloxytropane remained elusive. In this study, we successfully identified 3β-tigloyloxytropane synthase (TS), a BAHD acyltransferase, that catalyzes the formation of 3β-tigloyloxytropane. Considering the glycosidase-inhibiting activities of calystegines and their putative applications in the therapy of metabolic disorders, our finding provides a target for breeding calystegines-free *Solanaceae* crops and synthesizing valuable calystegines in plants or microbes.

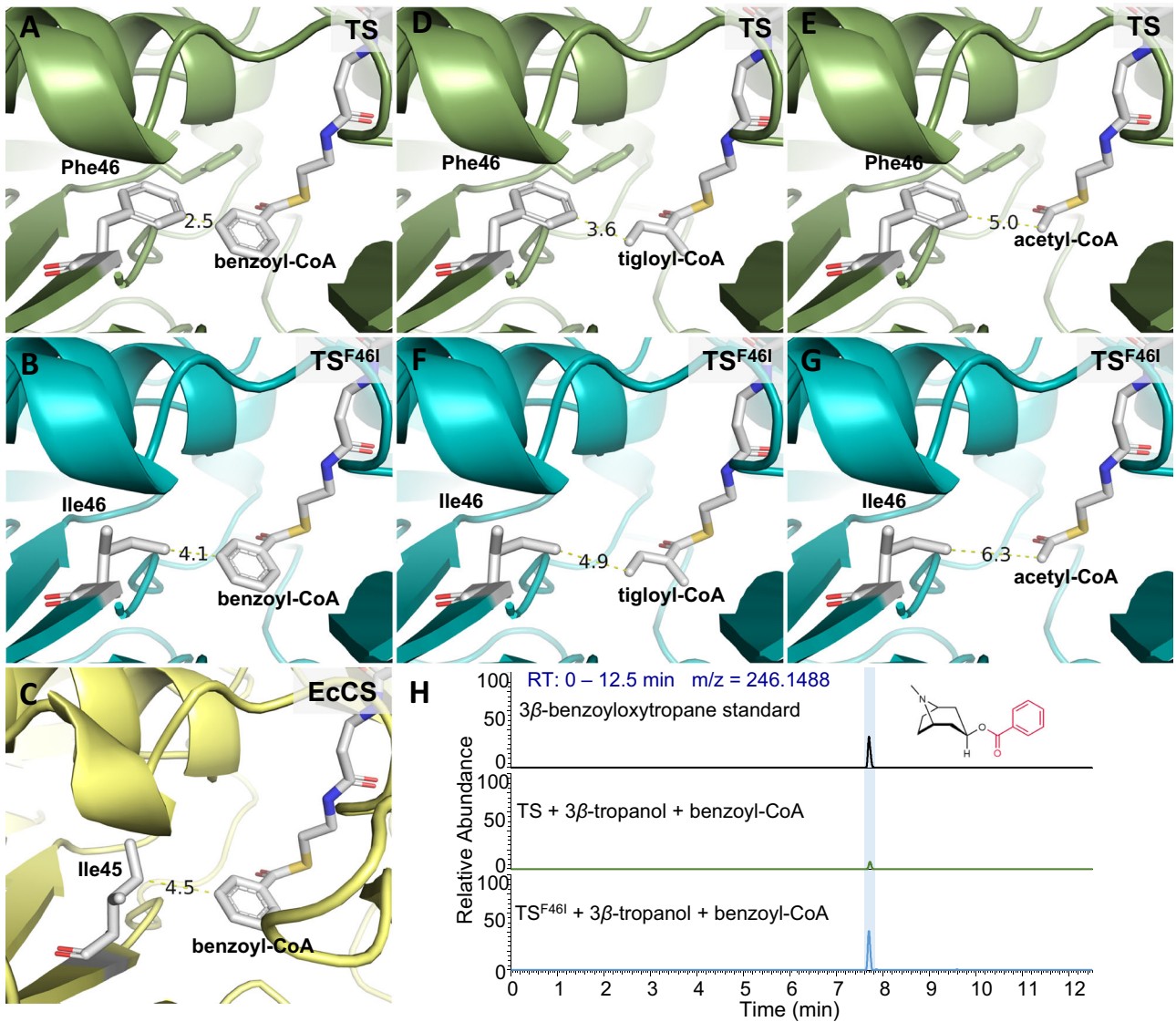

**Fig. 6 | Improvement in the substrate promiscuity of TS. A** Catalytic pocket of TS that contains benzoyl-CoA. **B** Catalytic pocket of TS$^{F46I}$ that contains benzoyl-CoA. **C** Catalytic pocket of EcCS that contains benzoyl-CoA. **D** Catalytic pocket of TS that contains tigloyl-CoA. **E** Catalytic pocket of TS that contains acetyl-CoA. **F** Catalytic pocket of TS$^{F46I}$ that contains tigloyl-CoA. **G** Catalytic pocket of TS$^{F46I}$ that contains acetyl-CoA. **H** Enzymatic assays of TS and TS$^{F46I}$ with benzoyl-CoA as the acyl donor.

Proper subcellular localization is essential for enzyme activity[62–64]. Although BAHD-ATs are generally localized in the cytosol[29], TS was localized in the mitochondria (Fig. 4). In particular, the 32 amino-acid in the N-terminal of TS plays an important role in TS activity, but it is sufficient but not essential for the mitochondrial localization of TS. The mitochondrial localization of TS was unexpected but reasonable. It is well established that tigloyl-CoA, a degradation product of iso-leucine, is produced in mitochondria[59,60,65,66]. The mitochondrial localization of TS suggests that this enzyme may utilize acyl donors in a rapid and economical manner. Earlier studies showed that mitochondria are a good organelle for the production of terpenes that require acetyl-CoA as a substrate[67,68]. Thus, a more efficient metabolic engineering strategy for BAHD-AT-mediated esters biosynthesis may be achieved by this mitochondrial localization feature of TS.

Based on the ternary complex model (Fig. 5A), we found that the acyl acceptor binding pocket of TS, composed of Ile35, Gln39, His162, Tyr280, Asn298, Leu300, and Trp340, stabilized 3β-tropanol in a suitable catalytic conformation (Fig. 5B). Combined with the enzymatic activity assays of TS mutants, we proposed the catalytic mechanism of TS. The His162 in the conserved HXXXD domain of BAHD-ATs is a general base that deprotonates the 3-hydroxyl of 3β-tropanol, priming it for nucleophilic attack on the carbonyl carbon of the acyl donor (Fig. 5E). In this process, CoA is then, released from the tetrahedral intermediate as a leaving group to produce the 3β-tropanol ester (Fig. 5E).

Like reported BAHD-ATs, TS exhibits acyl donor promiscuity. TS mainly catalyzed the formation of 3β-tigloyloxytropane, because its affinity for tigloyl-CoA is much higher than that for acetyl-CoA and benzoyl-CoA (Table 1). This may be due to the aromatic ring of benzoyl-CoA exhibiting greater steric hindrance compared to short-chain acyl-CoA thioesters[69]. For TS, Phe46 strictly controls the recognition of acyl donors through steric hindrance. When benzoyl-CoA was used as a substrate, the aromatic ring of Phe46 hinder the entry of benzoyl-CoA. Mutation of Phe46 to isoleucine decreased the steric hindrance significantly improved the utilization efficiency of benzoyl-CoA and greatly reduced the utilization efficiency of tigloyl-CoA (Fig. 6, Supplementary Fig. S35 and Table 1). Our findings on the catalytic mechanism of TS provided insights into the esterification mediated by BAHD-ATs and can be used to design diversity esters.

A highly efficient enzyme is key to the production of natural products by synthetic biology. A higher activity TS mutant, TS$^{S40T}$, was

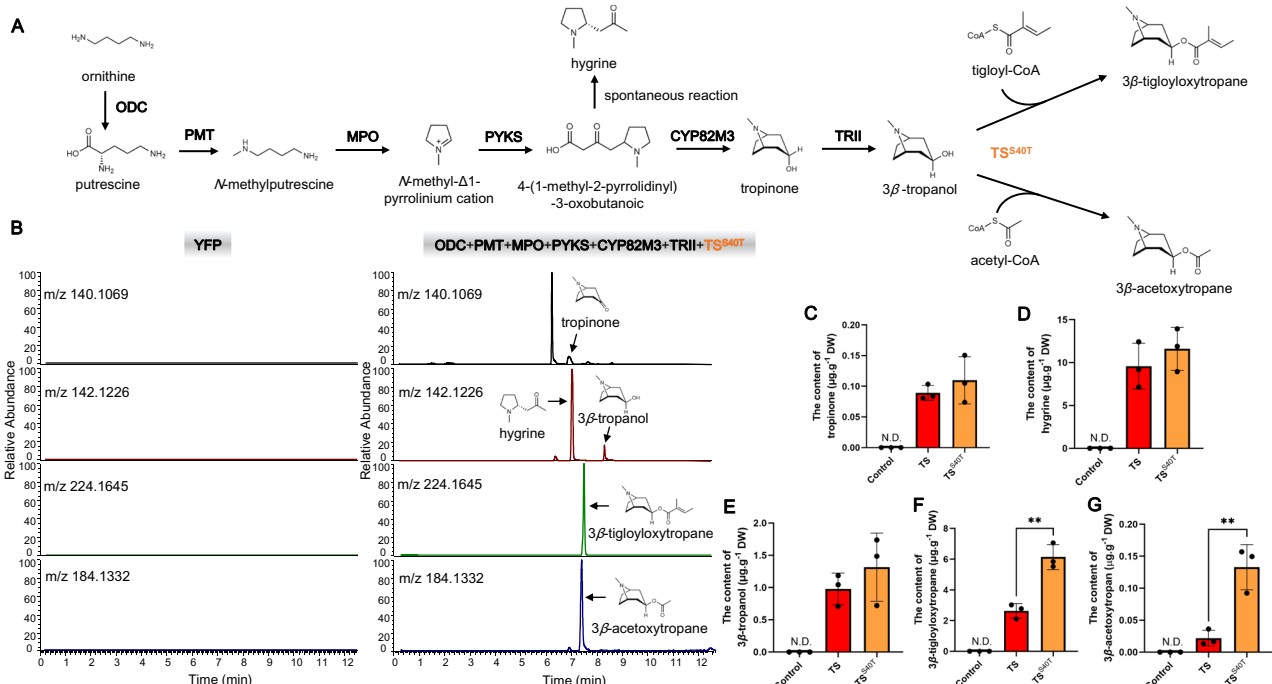

**Fig. 7 | De novo synthesis of 3β-tigloyloxytropane in *N. benthamiana*.**
**A** Reconstruction of the 3β-tigloyloxytropane biosynthetic pathway in *N. benthamiana*. **B** LC-MS detection of target metabolites in tobacco leaves. **C** The contents of tropinone. **D** The contents of hygrine. **E** The contents of 3β-tropanol. **F** The contents of 3β-tigloyloxytropane. **\*\*P = 0.003. G** The contents of 3β-acetoxytropane.

\*\*$P = 0.0068$. The control represents tobacco leaves expressing YFP. TS (TS^S40T) represents tobacco leaves coexpressing six TA genes and TS (TS^S40T). The data are presented as means values ± s.d. Leaves from three independent plants of each line were used for metabolite analysis. Statistical analysis was performed according to the two-sided independent sample *t*-test.

designed (Supplementary Fig. S35). Thr40 of TS^S40T formed a novel hydrogen bond with Asn135, which was not present in TS. These intramolecular interactions play an important role in enzyme stability by rigidifying the active center[70]. Thus, the hydrogen bond between Thr40 and Asn135 may contribute to the stability of the 3β-tropanol binding pocket, thereby improving the catalytic activity of the enzyme.

Consistent with the in vitro enzyme activity assays, the TS^S40T also exhibited higher esterification activity than TS in engineered tobacco, in which a de novo biosynthetic pathway for 3β-tropanol esters was constructed (Fig. 7). Although our results demonstrated the feasibility of synthesizing 3β-tigloyloxytropane from tobacco, the very low levels of 3β-tigloyloxytropane prevented the use of tobacco as an ideal platform for 3β-tigloyloxytropane production. Most of the metabolic flux was directed toward hygrine rather than towards 3β-tigloyloxytropane, and an insufficient supply of intermediates might limit 3β-tigloyloxytropane production (Fig. 7). To produce 3β-tigloyloxytropane more efficiently, we designed engineered strains of *E. coli* for the semi-biosynthesis of 3β-tigloyloxytropane. To overcome the limitation of lack of tigloyl-CoA in *E. coli* and improve the efficiency, we employed PcICS from *Pseudomonas chlororaphis*[61]. The advantages were obvious. On the one hand, using the substrate promiscuity of PcICS, synthesis of tigloyl-CoA from CoA and tiglic acid, increased the supply of tigloyl-CoA (Fig. 8C). On the other hand, CoA, the products of TS, can be used as substrates for PcICS (Fig. 8C). In this CoA recycling system, the expensive substrate, CoA, originated from *E. coli* itself and was continuously regenerated and utilized (Fig. 8C). Combined with the utilization of the highly efficient TS mutant, TS^S40T, we constructed a semi-biosynthesis 3β-tigloyloxytropane production system using only two readily available substrates, 3β-tropanol and tiglic acid and achieved 357.40 mg.L⁻¹ 3β-tigloyloxytropane production with 90.9% conversion rate (Fig. 8D).

To summarize, in this study, we identified the missing step in 3β-tigloyloxytropane formation by identifying the mitochondrion-localized BAHD acyltransferase 3β-tigloyloxytropane synthase (TS). We revealed the catalytic mechanism of TS and successfully improved its activity. Finally, we managed to reconstitute 3β-tigloyloxytropane biosynthesis in tobacco and *E. coli*. Our study helps characterize the enzymology and chemical diversity of TAs and provides an approach for producing 3β-tigloyloxytropane through synthetic biology.

## Methods
### Chemical synthesis
As commercial standards of 3β-tigloyloxytropane, 3β-acetoxytropane, 3β-benzoyloxytropane and tigloyl-CoA are not available, these three compounds were chemically synthesized in our laboratory. The synthetic methods used were described previously. The acids were converted to acyl chloride derivatives and then reacted with 3β-tropanol to prepare 3β-tropanol esters[71,72]. Tigloyl-CoA thioesters were synthesized from CoA and free acid via the catalysis of PyBOP[73].

To generate tigloyl chloride/acetyl chloride/benzoyl chloride: Tiglic acid/acetic acid/benzoic acid (10 mmol, 1 equivalent) was added portion-wise to oxalyl chloride (20 mmol, 1.50 equivalent), followed by one drop of N, N-dimethylformamide. After 2.5 h, the excess oxalyl chloride was removed under reduced pressure to provide tigloyl chloride/acetyl chloride as a light-yellow liquid.

3β-Tigloyloxytropane/3β-acetoxytropane/3β-benzoyloxytropane: 3β-tropanol (0.79 mmol, 1 equivalent) was dissolved in 5 ml of anhydrous tetrahydrofuran in a round-bottomed flask with a molecular sieve (0.4 nm), after which 4-dimethylaminopyridine (DMAP; 0.04 mmol) and triethylamine (1.58 mmol, 2 equivalent) were added. An inert atmosphere was established and maintained by a continuous flow rate of nitrogen. The mixture was cooled in an ice bath, after which tigloyl chloride/acetyl chloride (2.37 mmol, 3 equivalents) was introduced dropwise. The reaction mixture was left at 25 °C, under agitation, for 2 h, diluted with a saturated solution of Na₂CO₃ (10 ml) and distilled water (20 ml), and subsequently extracted with

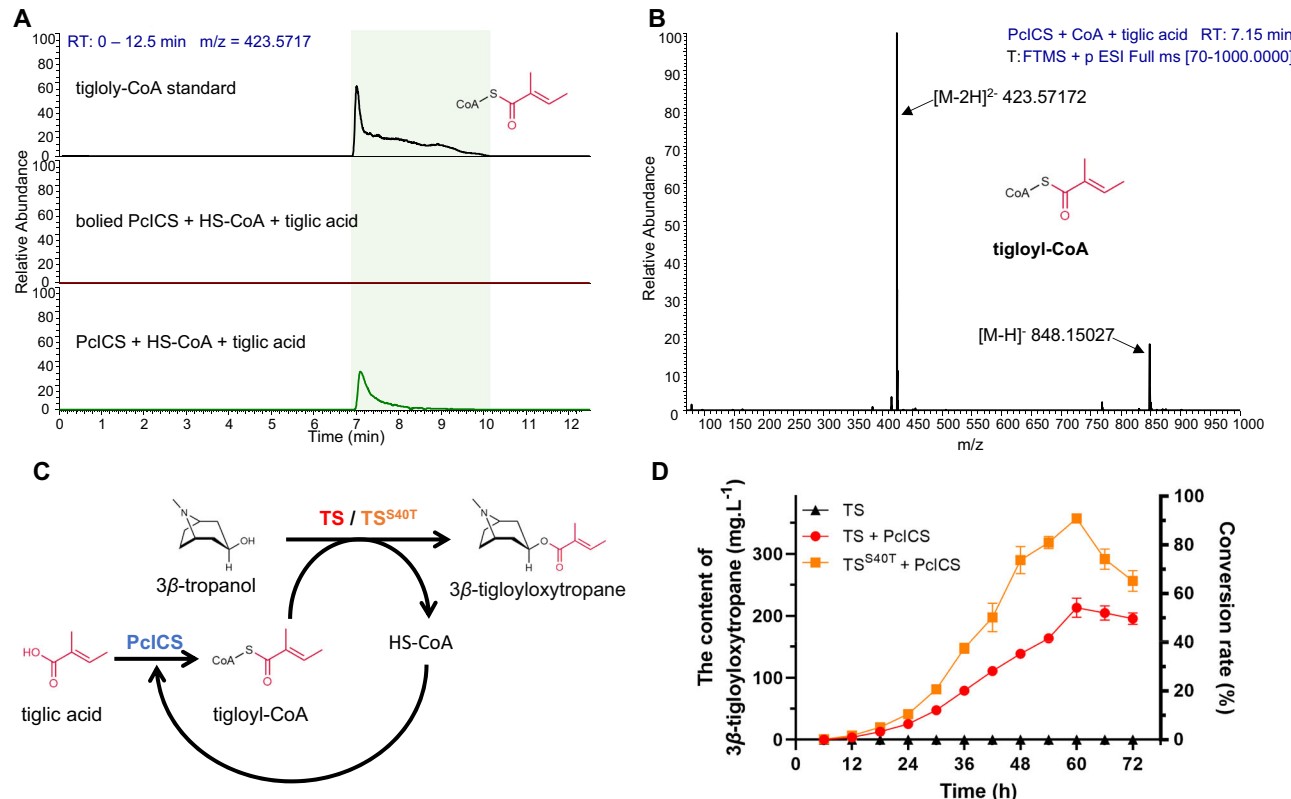

**Fig. 8 | *E. coli* bioreactors for the biosynthesis of 3β-tropanol esters. A** PcICS enzymatic assays. **B** MS data of tigloyl-CoA. **C** Semi-biosynthetic route of 3β-tigloyloxytropane. **D** Production of 3β-tigloyloxytropane. Left Y-axis, the content of 3β-tigloyloxytropane. Right Y-axis, the conversion rate from 3β-tropanol to 3β-tigloyloxytropane. HS-CoA means coenzyme A (CoA). Three independent transformants of each engineered *E. coli* were used in the semi-biosynthesis of 3β-tigloyloxytropane. The data are presented as means values ± s.d.

chloroform (3 × 30 ml). The organic portion was dried with anhydrous sodium sulfate and concentrated under reduced pressure at 35 °C. Then, the product was further purified by thin-layer chromatography and high-performance liquid chromatography (HPLC).

Tigloyl-CoA: Tiglic acid (50 μmol, 5.0 equivalent), PyBOP (16 μmol, 1.6 equivalent), and $K_2CO_3$ (40 μmol, 4.0 equivalent) were dissolved in 3 mL of freshly distilled tetrahydrofuran (THF) under argon. CoA (10 μmol, 1.0 equivalent) was dissolved in 1 mL of $H_2O$ ($O_2$ was removed by sonication) and added dropwise to the THF solution. After stirring at 25 °C for 3 h, the reaction mixture was directly subjected to a HPLC on C18 semi-preparative reversed-phase column.

The fragment ions were analyzed through high-resolution tandem mass spectrometry (MS/MS) analysis to preliminarily determine their structures (Supplementary Fig. S1–S4). Nuclear magnetic resonance (NMR), specifically, $^1H$-NMR, $^{13}C$-NMR, and 2D-NMR, was used to further confirm the structures of the synthesized compounds (Supplementary Fig. S5–S24 and Supplementary Data 1).

### Bioinformatics analysis

Hidden Markov Model (HMM) for BAHD-ATs (PF02458) was used to identify the BAHD genes through the hmmsearch program against *A. belladonna* transcriptomes from the Medicinal Plant Genomics Resource (http://mpgr.uga.edu/) in HMMER 3.3.2[74]. Batch CD-Search (https://www.ncbi.nlm.nih.gov/Structure/bwrpsb/bwrpsb.cgi) was used to further confirm that the candidate genes belonged to the BAHD-AT family. TBtools-II v2.042 was used to construct a heatmap of the association analysis of the metabolites and BAHD gene family expression patterns[75]. Amino acid sequence alignment was performed using the E-INS-I method in MAFFT v7.475. The phylogenetic tree was constructed using IQ-TREE v2.1.2[76] with the maximum likelihood

model (LG + F + G4). The tree was rooted using the algal enzyme clade (clade 0) and classified according to the previous method[35]. In addition, the tree was annotated and visualized with iTOL v6[77].

### Expression analysis of *TS*

Different organs, including the secondary roots, primary roots, mature seeds, ripe fruits, flowers, stems, green fruits, and leaves, were harvested from 4-month-old *A. belladonna* plants; these organs were used for RNA isolation and cDNA synthesis via kits from TIANGEN Biotech (Beijing, China). Real-time quantitative PCR was used to analyze the tissue profile of TS using an iQ5 system (Bio-Rad, USA), with *PGK* serving reference gene[78]. Amplifications were carried out using SYBR Green PCR MasterMix (Novoprotein, Shanghai, China). The primers used for qPCR are listed in Supplementary Table S1.

### Purification of MBP-tagged TS and enzymatic assay

To obtain sufficient protein for biochemical characterization, the coding sequence of TS was synthesized according to the codon usage bias of *E. coli*, forming a codon-optimized version designated TSopt. The coding sequence of TSopt was amplified by a pair of primers with BamHI and PstI restriction sites and then inserted into pMAL-c5x to generate the prokaryotic expression plasmid, pMAL-c5x-TS. The primers used to construct the vectors are listed in Supplementary Table S1. The pMAL-c5x-TS plasmid was subsequently introduced into *E. coli* BL21 (DE3) for expression. Protein expression was induced overnight at 16 °C with 0.25 mM IPTG in LB medium. The MBP-tagged TS protein was purified using amylose resin (Smart-Lifesciences). After desalting, fresh proteins were immediately subjected to enzymatic assays. 3β-Tropanol was combined with three different acyl donors (tigloyl-CoA, benzoyl-CoA, and acetyl-CoA) to form a substrate for

three groups of enzymatic assays. The reaction mixture was 50 mM glycine-NaOH buffer containing 20 μg of purified TS, 1 mM 3β-tropanol and tigloyl-CoA (or benzoyl-CoA or acetyl-CoA).

To explore the optimal reaction conditions of TS, a pH range from 6 to 10.6 and a temperature range from 20 °C to 50 °C were tested with tigloyl-CoA and 3β-tropanol. Finally, the enzyme kinetics were measured at pH 8.6 and 30 °C. Various concentrations of 3β-tropanol (0.01–2 mM) and tigloyl-CoA (or benzoyl-CoA, acetyl-CoA) (0.01–1 mM) were used for the analysis of enzyme kinetics.

### Subcellular localization analysis
Subcellular localization prediction of TS was performed using TargetP-2.0 (https://services.healthtech.dtu.dk/services/TargetP-2.0/)[79]. The coding sequence of TS, 32 amino acids at its N-terminus, and TS without 32 amino acids at its N-terminus were amplified by a pair of primers with *Bgl*II and *Hind*III restriction sites and then inserted into the plant expression vector pGD3G-YFP to generate pGD3G-TS-YFP, pGD3G-N32-YFP, and pGD3G- TS$^{Del-N32}$-YFP. The primers used for vector construction are listed in Supplementary Table S1. Subsequently, the constructs were subsequently transferred into the *A. tumefaciens* strain GV3101, which was subsequently cotransformed into *N. benthamiana* leaves. After 48 h of cultivation, the leaves were prepared into protoplasts and treated with MitoTracker Red (Beyotime, Shanghai, China). The Confocal microscopy (Olympus FV1200) was used to analyze the subcellular localization of TS. Red fluorescence was emitted from the MitoTracker Red. Yellow fluorescence was emitted from TS-YFP, N32-YFP, and TS$^{Del-N32}$-YFP.

To further determine the localization of TS in *A. belladonna*, mitochondria were extracted from the roots of 2-month-old *A. belladonna* plants. After differential centrifugation was performed to separate mitochondria[52], the following steps were performed: I. Ten grams of fresh roots were frozen in liquid nitrogen and ground thoroughly into a fine powder. II. Add Mitochondrial extraction buffer (70 mM sucrose, 210 mM mannitol, 10 mM HEPES, 1 mM EDTA, 1 mM PMSF, pH 7.5) was added, and the samples were homogenized with a Dounce tissue grinder for 10 cycles. III. The homogenate obtained in the previous step was centrifuged at $600 \times g$ for 10 min at 4 °C, after which the supernatant was collected. IV. Centrifuge at $1200 \times g$ for 10 min at 4 °C and collect the supernatant. V. Centrifuge at $11,000 \times g$ for 20 min at 4 °C. The supernatant obtained at this stage was the cytosol, and the pellet was the mitochondria. Western blotting was also conducted to further determine the mitochondria and cytosol content. Western blotting was performed with an anti-VDAC1 antibody (rabbit, 1:2000 dilution), which was used as a mitochondrial marker[52–54]. The anti-VDAC1 antibody was purchased from Orizymes with Catalog Number PAB220312. Goat anti-rabbit IgG antibody (goat, 1:5000 dilution) was used as the secondary antibody. The Goat anti-rabbit IgG antibody was purchased from Biosharp with Catalog Number BL003A. The chemiluminescent signals were detected using a Tanon 5200 Fully Automatic Chemiluminescence Image Analysis System (TANON, Shanghai, China). Next, the cytosol and mitochondria were treated using ultrasound (20 kHz, ultrasound for 3 s, stop for 7 s, 50 cycles) and the protein concentration was measured using the BCA method. Then, the crude protein from the mitochondria/cytosol was incubated with the substrate for the enzymatic assay. The reaction mixture was 50 mM glycine-NaOH buffer (pH 8.6) containing 50 μg of crude protein from mitochondria/cytosol, 1 mM 3β-tropanol, and tigloyl-CoA.

### Virus-induced gene silencing (VIGS)
A 538 bp TS fragment was inserted into the tobacco rattle virus (TRV) vector (pTRV2) using the restriction enzymes XhoI and KpnI to construct the plasmid pTRV2-TS. The primers used for vector construction are listed in Supplementary Table S1. Then, pTRV1 and pTRV2-TS were co-introduced into *Agrobacterium tumefaciens* GV3101, which was subsequently transiently transformed into 18-day-old *A. belladonna* cotyledons. The pTRV1 and pTRV2 plasmids were co-introduced as control groups. After 28 days, the roots were subjected to expression and metabolite analysis. The operation process of VIGS that was used in the present study was previously reported[80].

### Overexpression of TS in hairy root cultures of *A. belladonna*
The *TS* coding sequence was amplified by a pair of primers with *Bam*HI and *Sac*I restriction sites and then inserted into pBI121 to generate the plant expression vector, pBI121-TS. The primers used for vector construction are listed in Supplementary Table S1. To evaluate the effects of TS overexpression on the biosynthesis of TAs, we established root cultures of *A. belladonna* through *Agrobacterium*-mediated transformation. Root cultures of *A. belladonna* were established according to methods already described[4,30,81]. The root cultures were grown in a flask containing 100 mL of Murashige and Skoog liquid media and harvested after they had been cultured for 28 days at 25 °C in darkness. Eight root lines, independently transformed with pBI121, were used as control root cultures (CK group). Four lines with significantly increased expression of target genes (OE-1, OE-2, OE-3, and OE-4), independently transformed by the corresponding engineered vectors, were randomly selected for metabolite analysis.

### Metabolite analysis
Freeze-dried root cultures, including those of the roots and hairy roots of *A. belladonna* and tobacco leaves, were ground into fine powder and used for alkaloid extraction. The extraction and detection of these TAs were based on a previously reported methodology[82]. In brief, 250 mg of dry root powder was placed in 1 ml of extraction buffer (20% methanol, containing 0.1% formic acid) and shaken for 2 h at 200 rpm and 25 °C. Each extract was passed through a 0.22 μm Nylon 66 filter (Jinteng, Tianjin, China), and the filtrate was subsequently diluted 100-fold for future analysis. The alkaloid content in the extract was analyzed by an Orbitrap Exploris 120 LC–MS (Thermo Scientific, Pittsburgh, PA, USA). Measurements were performed using electron spray ionization (ESI) in positive ion mode and full MS mode. The instrument parameters were set as follows: sheath gas flow rate of 35, aux gas flow rate of 10, spray voltage of 3.00 kV, capillary temperature of 350 °C, S-lens RF level of 50, aux gas heater temperature of 350 °C. The level of High Energy Collision Dissociation was 30 in MS/MS analysis.

3β-Acetoxytropane, hygrine, tropinone, 3α-tropanol, and 3β-tropanol were analyzed using a CORTECS UPLC HILIC column (2.1 mm × 100 mm, 1.6 μm) obtained from Thermo Scientific (Pittsburgh, PA, USA). The system's flow rate was 0.3 ml.min$^{-1}$, and the temperature was 35 °C. The sample solution per injection was 3 μl; the elution procedures are described in Supplementary Table S2.

3β-Tigloyloxytropane, 3β-benzoyloxytropane, littorine, hyoscyamine, anisodamine, scopolamine, tropanol hexosides, tigloyl norpseudotropine, tigloyl 1-hydroxynorpseudotropine and calystegines A3 were analyzed with a Hypersil GOLD C18 column (2.1 mm × 100 mm, 1.9 μm) obtained from Thermo Scientific (Pittsburgh, PA, USA), for which the sample solution per injection was 3 μl. The system's flow rate was 0.3 ml.min$^{-1}$, and the temperature was 35 °C. The corresponding elution procedures are detailed in Supplementary Table S3.

### Molecular simulation
AlphaFold2 v2.3.0 was used for building the protein model of TS and predicting the substrate pocket[83]. The ligands (3β-tropanol, acetyl-CoA, tigloyl-CoA and benzoyl-CoA) were downloaded from the PubChem database (https://pubchem.ncbi.nlm.nih.gov/) and docked into the cofactor-binding site of TS using AutoDock Tools v1.5.6[84]. Then, independent docking runs for different substrates with TS were generated and complex structures with lower binding energies and favorable orientations were selected. The interactions between the substrates and TS were analyzed using PLIP v2.3.0[85]. PyMOL 2.1 (http://www.pymol.org) was used to view the molecular interactions and process the image.

## Consensus protein design

Consensus protein design is based on the hypothesis that at a given position, compared with nonconserved amino acids, the respective consensus amino acid contributes more than average to the stability of the protein[86]. Through BLASTP from NCBI server, genes with more than 50% identity to the TS amino acid sequence from the public database were obtained for consensus protein design. Conservative substitution analysis was performed on residues within the 5 Å range centered on His162 in TS. The seqlogo drawn by TBtools-II v2.042 displayed conservative substitutions[75].

## Biosynthesis of 3β-tigloyloxytropane in *N. benthamiana*

For the de novo synthesis of 3β-tropanol esters in *N. benthamiana*, EnODC, AbPMT, AbMPO, AbPYKS, AbCYP82M3, DsTRII, and AbTS (or its mutants) were coexpressed in tobacco leaves. The coding sequences of these genes were amplified by a pair of primers with AgeI and XhoI restriction sites and then inserted into pEAQ-HT to generate a series of transient expression vectors. The primers used for vector construction are listed in Supplementary Table S1. These transient expression vectors were subsequently introduced into *Agrobacterium tumefaciens* GV3101 and transiently transformed into tobacco leaves. After 5 days of cultivation, the leaves were harvested and subjected to metabolite analysis.

In the semi-synthetic experiment of tobacco, PcICS optimized according to tobacco codons was constructed into pEAQ-HT. Then, AbTS (or its mutants) and PcICS were expressed in the tobacco leaves via a process mediated by GV3101. 2 Days later, tobacco leaves were fed 3β-tropanol and tiglic acid at a concentration of 1 mM. After 3 days of cultivation, the leaves were harvested and subjected to metabolite analysis.

## Biosynthesis of 3β-tigloyloxytropane in *E. coli*

The coding sequences of TS (or its mutants) and PcICS were inserted into pETDuet-1 to generate a series of prokaryotic expression vectors. The primers used for vector construction are listed in Supplementary Table S1. Then, these prokaryotic expression vectors were introduced into *E. coli* (BL21). The engineered *E. coli* was cultured in 100 ml of LB liquid medium to an OD600 of 0.6. Next, IPTG was added to a final concentration of 0.25 mM and substrate of 250 mg.L$^{-1}$ (tiglic acid and 3β-tropanol) of substrate was added to the culture medium. One milliliter of culture solution was added every 6 h for product content analysis.

## Statistics and reproducibility

GraphPad Prism 8 software was used for regular statistical analysis and enzyme kinetic analysis. A two-tailed Student's *t*-test was used to calculate significant differences among samples or genotypes. Details of biological replicates used in various experiments are provided in the "Methods" section as well as in the main figures and Supplementary Figs. legends, wherever necessary. No statistical method was used to predetermine the sample size. No data were excluded from the analyses. Four lines with significantly increased expression of target genes (OE-1, OE-2, OE-3, and OE-4), independently transformed by the corresponding engineered vectors, were randomly selected for metabolite analysis.

## Reporting summary

Further information on research design is available in the Nature Portfolio Reporting Summary linked to this article.

## Data availability

The data used for transcriptome analysis comes from Medical Plant Genomics Resources (http://mpgr.uga.edu/). The sequences (46 BAHD acyltransferases of *A. belladonna* and 196 functionally identified BAHD acyltransferases) used to construct the phylogenetic tree are listed in Supplementary Data 1. Phylogenetic analysis of BAHD acyltransferases

is shown in Supplementary Data 2. The $^1$H-NMR, $^{13}$C-NMR, and 2D-NMR data of the chemicals synthesized in this study are listed in Supplementary Data 3. The coding sequence of TS has been uploaded to NCBI, and its accession number is OP677554. Source data are provided with this paper.

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

## Acknowledgements

This work was financially supported by the National Natural Science Foundation of China (32370277), STI2030—Major Projects (2023ZD04076), the Forth National Survey of Traditional Chinese Medicine Resources, Chinese or Tibet Medicinal Resources Investigation in Tibet Autonomous Region (State Administration of Chinese Traditional Medicine 20191217-540124 and 20200501-542329), Experimental Technology Research Project of Southwest University (SYJ2020004), Natural Science Foundation of Chongqing (cstc2021jcyj-msxmX0571), and Key Project at Central Government Level: The Ability Establishment of Sustainable Use for Valuable Chinese Medicine Resources (2060302). We thank Dr. Hsihua Wang (the Mass Spectrometry Core Facility in College of Life Sciences, Sichuan University) for the assistance in the metabolic analysis.

## Author contributions

Zhihua Liao, Junlan Zeng, Kexuan Tang, Fangyuan Zhang designed the experiments and wrote the paper; Junlan Zeng, Xiaoqiang Liu, Fei Qiu, Fangyuan Zhang, Zhaoyue Dong, Mingyu Zhong, Tengfei Zhao and Hongbo Zhang conducted the experiments; Chunxian Yang maintained plant materials; Lingjiang Zeng conducted LC-MS analysis; Zhaoyue Dong, Mingyu Zhong and Min Chen prepared and identified chemicals; Fangyuan Zhang, Junhui Zhou and Xiaozhong Lan performed statistical analysis. Zhihua Liao provided resources supporting this study.

## Competing interests

On March 1, 2024, J.Z and Z.L., as inventors, applied for patents on the application methods of TS, TS$^{S40T}$, and PcICS in 3β-tigloyloxytropane production (CN2024102341305, filed by Integrative Science Center of Germplasm Creation in Western China (CHONGQING) Science City). On March 1, 2024, J.Z. and Z.L., as inventors, applied for patents on the application methods of TS$^{F46I}$ and TS$^{S40T-F46I}$ in 3β-benzoyloxytropane production (CN2024102341305, Integrative Science Center of Germplasm Creation in Western China (CHONGQING) Science City). The remaining authors declare no competing interests.
