## [Peer Review File · Nature Communications]

REVIEWER COMMENTS

Reviewer #1 (Remarks to the Author):

The manuscript by Zheng et al. described *Atropa belladonna* mitochondrion localized BAHD acyltransferase involved in calystegine biosynthesis. Authors reported root-specific BAHD acyltransferase, 3 β -Tigloyloxytropine Synthase aka TS from *Atropa belladonna* that is responsible for the formation of 3 β -tigloyloxytropine (tigloidine), the key intermediate in calystegine biosynthesis and potential drug treating neurodegenerative disease. The work has clear strengths. The range of experiments is impressive, including functional analysis in planta, in vitro, molecular docking and mutational studies etc.

There are several cosmetic problems with the manuscript. This is a very difficult manuscript to read and review. The text needs editing from a professional and the data presentation and reference to the presented data has to be improved before it is accepted for high standard journal like Nature communication. Several typos and inconsistent phrases need special attention. Methods section is incomplete while missing important details. Below are my comments that can be used to improve the manuscript.

Introduction provides a lot of detail, which is hard to follow. What is the big question regarding the enzymatic step under investigation? Introduction should be clear and succinct.

Line 76-78: It would be helpful if authors provide recent references for use of tigloidine or tropigline as substitute for atropine in diseases treatment. The reference provided are from 1950's and 60's.

Line 123: we firstly, change to 'first'

Line 129: please provide full name for abbreviation HMMER

Line 132-134: Figure panels are mislabeled, Fig. 2C, 2D etc.

Line 135: The phylogenetic tree supports that TS recognizes the 3 β -tropane ring skeleton. Please change the subheading. It is misleading. In addition, it would be nice to mark EcCS and EcBAHD8 in the phylogeny Figure 2D.

Line 197: BAHD-AT family proteins generally exhibit substrate promiscuity. Please provide relevant references.

Line 205-206: remove 'obviously'

Supplementary Figure 3: panel A, it should be 3 β -benzoyloxytropine standard and not 3 β -tigloyloxytropine. Also its 'standard' and not 'standards'

Line 209-219: VIGS results were surprising with respect to no reports on accumulation of pseudotropine or its downstream hexosides? What about acylated pseudotropines that could be formed by other BAHD activities?

Figure 3D-H and Supplementary Figure 4: why the scale of y-axis is inconsistent? Keep uniformity while presenting the data. Check when referring Supplementary Figures, for example, I see Supplementary Fig. 3 at one place, and Supplementary Fig. S4 at another place.

Line 217-219: These gene-silencing results confirmed the enzymatic assay results that TS was responsible for the formation of 3 β -tigloyloxytropine, and indicated that TS was involved in the biosynthesis of calystegine A3, the major compound of calystegines. Please rephrase sentences, as gene silencing did not confirm the enzyme assay results. Both in planta and in vitro assays showed that TS catalyzes the formation of 3 β -tigloyloxytropine in the biosynthetic pathway of calystegine A3

Line 221-230: Again quite surprising with overexpression studies as only one metabolite is accumulated and there is no change in downstream or competitive pathway branches.

Figure 4A: is it N-32 or N-43?

Line 231-248: If TS has mitochondrial targeting signal, did authors remove this sequence for E. coli expression? How did authors get soluble TS protein? I did not see this information in relevant results section? Also authors should do localization of TS without these 32 amino acids in tobacco to support their claims.

Figure 7 can be moved to supplementary. It's quite difficult to follow all these mutational studies.

Line 409-419: Did authors infiltrate tigloyl-coA as a substrate after agro-infiltration? I wonder whether N. benthamiana produce tigloyl CoA? I did not see this information anywhere in manuscript. Did authors try to feed tropinone, which is extremely cheap substrate and can result in enhancement of tigloidine instead of infiltrating 7-8 enzymes?? Authors should also include version of TS without mitochondrial signal in benthamiana assay and test whether tigloidine is formed or not? This will provide the information where sub-cellular localization of TS is essential for activity or not?

The discussion is very long and it's quite repetitive explaining the results. I suggest reducing it significantly. What are the key messages that one need to understand to appreciate the context and importance of this study?

Reviewer #2 (Remarks to the Author):

In the present manuscript, Zhihua Liao and co-workers reported a unusual mitochondria localized BAHD acyltransferases (TS) from *Atropa belladonna*, which catalyzed the esterification of 3 β -tropanol to form

3 β -tigloyloxytropone specifically. The function of TS was further confirmed in planta by gene silence and overexpression. The catalytic mechanism was discussed by molecular docking, which enable to generate a mutant (TSS40T) with higher activity. They also successfully reconstituted the pathway in tobacco and *E. coli*, which provide a basis on 3 β -tigloyloxytropone production via metabolic engineering. The authors should address the following minor comments and concerns below before it could be published in Nature communications:

1.TS showed substrate promiscuity in vitro. Did the authors see any other acylated products when the whole pathway was reconstituted in the *Nicotiana benthamiana*, which can provide various acyl donors in vivo.

2.Heterologous expression system always cannot provide sufficient amount of cofactors for biosynthetic genes from other organisms. In *E. coli*, this problem was solved by introducing PciCS. The authors should also check the reactivity of PciCS in tobacco to see whether it can boost the production of 3 β -tigloyloxytropone when tiglic acid is infiltrated.

3.The reaction mechanism in Figure 5e is not correct. The arrow pushing should begin from NH in His162 and end with the protonation of -SCoA.

4.The phylogenetic tree in Figure 2d and chemical structures in Figure 2e are too small. It better to show these two in a new figure.

5.Ref3 should be Nat. Commun. 13, 4994 (2022)

6.Ref10 should be Stem Cell Res. Ther. 10, 178 (2019)

7.Ref17 should be Nat. Commun. 13, 3832 (2022)

8.Ref27 should be Proc. Natl. Acad. Sci. U. S. A. 119, e2215372119 (2022)

9.Ref31 should be Proc. Natl. Acad. Sci. U. S. A. 120, e2302448120 (2023).

10.Ref41 should be Nat. Commun. 12, 1563 (2021).

11.Ref42 should be Plant Methods 16, 149 (2020)

12.Ref43 should be J. Pineal Res. 63, e12429 (2017).

13.Ref62 should be Nat. Commun. 14, 597 (2023).

14.Ref63 should be Planta 252, 6 (2020).

15.Ref64 should be Sci.Rep. 6, 33797, (2016).

16.Ref69 should be BMC Bioinformatics 11, 431 (2010)

17.Ref77 should be Nat. Commun. 9, 5281 (2018)

REVIEWER COMMENTS

Reviewer #1 (Remarks to the Author):

**1. Comment:** The manuscript by Zeng et al. described *Atropa belladonna* mitochondrion localized
BAHD acyltransferase involved in calystegine biosynthesis. Authors reported root-specific BAHD
acyltransferase, 3 β -Tigloyloxytropine Synthase aka TS from *Atropa belladonna* that is responsible for
the formation of 3 β -tigloyloxytropine (tigloidine), the key intermediate in calystegine biosynthesis and
potential drug treating neurodegenerative disease. The work has clear strengths. The range of
experiments is impressive, including functional analysis in planta, in vitro, molecular docking and
mutational studies etc.

There are several cosmetic problems with the manuscript. This is a very difficult manuscript to read
and review. The text needs editing from a professional and the data presentation and reference to the
presented data has to be improved before it is accepted for high standard journal like Nature
communication. Several typos and inconsistent phrases need special attention. Methods section is
incomplete while missing important details. Below are my comments that can be used to improve the
manuscript.

**Response:** Thank you very much for your constructive comments about the paper. The language was
edited by an expert from Nature Research Editing Service. You can see this in the revised manuscript.
We also revised Methods section with details.

**2. Comment:** Introduction provides a lot of detail, which is hard to follow. What is the big question
regarding the enzymatic step under investigation? Introduction should be clear and succinct.

**Response:** Thank you very much for your professional comments. We have rewritten and polished the
Introduction section, and the changes are shown in the revised manuscript. Briefly, 1) we provided
background information on the known biosynthetic pathways of hyoscyamine, scopolamine, and
cocaine and highlighted the limited understanding of the biosynthesis of 3 β -tropanol derivatives. 2)
Subsequently, we presented our aims, which are to fill this gap by elucidating the genes and enzymes
responsible for the formation of 3 β -tropanol esters, with a particular focus on 3 β -tigloyloxytropine. 3)
Because esterification is a crucial chemical modification in plant metabolism, we introduced two

enzyme families, serine carboxypeptidase-like acyltransferases (SCPL-ATs) and BAHD
acyltransferases (BAHD-ATs), which play key roles in metabolite esterification. 4) We also highlighted
the subcellular compartmentation of these enzymes, emphasizing the unique localization of SCPL-ATs
in vacuoles and the generally cytosolic presence of BAHD-ATs. 5) Finally, we outlined the main
objective of this study.

**3. Comment:** Line 76-78: It would be helpful if authors provide recent references for use of tigloidine
or tropigline as substitute for atropine in diseases treatment. The reference provided are from 1950's
and 60's.

**Response:** Tigloidine is listed on page 1028 of the "Index Nominum 2000: International Drug
Directory". We added it to the reference section.

**4. Comment:** Line 123: we firstly, change to 'first'

**Response:** We corrected this.

**5. Comment:** Line 129: please provide full name for abbreviation HMMER

**Response:** Thank you very much for this suggestion. HMMER is a tool used to search sequence
databases for sequence homologues and perform sequence alignments using probabilistic models called
profile hidden Markov models (profile HMMs). HMMER does not have a full name
(<http://hmmer.org/>).

**6. Comment:** Line 132-134: Figure panels are mislabeled, Fig. 2C, 2D etc.

**Response:** We corrected them.

**7. Comment:** Line 135: The phylogenetic tree supports that TS recognizes the 3β -tropane ring skeleton.
Please change the subheading. It is misleading. In addition, it would be nice to mark EcCS and
EcBAHD8 in the phylogeny Fig. 2D.

**Response:** We have changed this subheading to "Phylogenetic analysis of BAHD acyltransferases". In
addition, EcCS and EcBAHD8 are highlighted by blue font in Fig. 2D.

**8. Comment:** Line 197: BAHD-AT family proteins generally exhibit substrate promiscuity. Please
provide relevant references.

**Response:** We added relevant references to the first sentence of this paragraph. The relevant references
are listed below:

Walker, A. M. *et al.* Elucidation of the structure and reaction mechanism of sorghum
hydroxycinnamoyltransferase and its structural relationship to other coenzyme a-dependent
transferases and synthases. *Plant Physiol.* **162**, 640–651 (2013).

Levsh, O. *et al.* Dynamic Conformational States Dictate Selectivity toward the Native Substrate in a
Substrate-Permissive Acyltransferase. *Biochemistry* **55**, 6314–6326 (2016).

Liu, X. *et al.* Crystal structure of the plant feruloyl–coenzyme A monolignol transferase provides
insights into the formation of monolignol ferulate conjugates. *Biochem. Biophys. Res. Commun.*
**594**, 8–14 (2022).

Kim, C. Y. *et al.* Emergence of a proton exchange-based isomerization and lactonization mechanism in

the plant coumarin synthase COSY. *Nat. Commun.* **14**, (2023).

**9. Comment:** Line 205-206: remove ‘obviously’

**Response:** We removed this term.

**10. Comment:** Supplementary Figure 3: panel A, it should be 3β -benzoyloxytropane standard and not
3β -tigloyloxytropane. Also its ‘standard’ and not ‘standards’

**Response:** We corrected them.

**11. Comment:** Line 209-219: VIGS results were surprising with respect to no reports on accumulation
of pseudotropine or its downstream hexosides? What about acylated pseudotropines that could be
formed by other BAHD activities?

**Response:** According to your comments, we reanalyzed the LC-MS data. Compared with those in the
control lines, the 3β -tropanol, 3α -tropanol, and tropanol hexoside contents in the TS-silenced lines
significantly increased (Fig. 3D and Supplementary Fig. S4).

**Fig. 3. Functional identification of TS.** (A) TS enzymatic assays with tigloyl-CoA as the acyl donor
and 3β -tropanol as the acyl acceptor. (B) Mass spectrometry (MS) data of 3β -tigloyloxytropane. (C)
Relative expression levels of *TS* in *A. belladonna* seedlings. (D) Contents of 3β -tropanol in *A.*
*belladonna* seedlings. (E) Contents of 3β -tigloyloxytropane in *A. belladonna* seedlings. (F) Contents of
3β -acetoxytropane in *A. belladonna* seedlings. (G) Contents of tigloyl norpseudotropine in *A.*
*belladonna* seedlings. (H) The contents of tigloyl 1-hydroxynorpseudotropine in *A. belladonna*
seedlings. (I) Contents of calystegine A3 in *A. belladonna* seedlings. Control, control line obtained by
empty plasmid transformation. VIGS-TS, TS silenced line. Fifteen independent plants were used in the
VIGS assays. (J) Relative expression levels of *TS* in *A. belladonna* hairy root cultures. (K) Contents of
3β -tropanol in *A. belladonna* hairy root cultures. (L) Contents of 3β -tigloyloxytropane in *A. belladonna*
hairy root cultures. (M) Contents of 3β -acetoxytropane in *A. belladonna* hairy root cultures. (N)
Contents of tigloyl norpseudotropine in *A. belladonna* hairy root cultures. (O) Contents of tigloyl

1-hydroxynorpseudotropine in *A. belladonna* hairy root cultures. (P) Contents of calystegine A3 in *A.*
 *belladonna* hairy root cultures. CK, eight independently transformed root culture lines transformed
 with pBI121. OE denotes all independently transformed root culture lines overexpressing TS (each line
 contained three biological replicates), including OE-1, OE-2, OE-3, and OE-4. DW, dry weight. The
 bars indicate the mean \pm standard error. DW, dry weight. ** indicates a significant difference from the
 CK /Control) group at $P < 0.01$, according to the two-sided independent sample *t*-test.

**Supplementary Fig. S4. The influence of silencing TS on the content of compounds involved in**
 **the competitive metabolic flow of TS in *A. belladonna* seedlings.** (A) 3 α -Tropanol. (B) Tropanol
 hexoside. (C) Littorine. (D) Hyoscyamine. (E) Scopolamine. ** indicates a significant difference from
 the CK group at $P < 0.01$ according to the two-sided independent sample *t*-test.

**12. Comment:** Figure 3D-H and Supplementary Figure 4: why the scale of y-axis is inconsistent? Keep
 uniformity while presenting the data. Check when referring Supplementary Figures, for example, I see
 Supplementary Fig. 3 at one place, and Supplementary Fig. S4 at another place.

**Response:** To visualize the difference between these data, different scales were used on the y-axis, and
 the unit of $\mu\text{g}\cdot\text{g}^{-1}$ DW was used only in the revised figures. We checked the supplementary figures and
 revised them accordingly.

**13. Comment:** Line 217-219: These gene-silencing results confirmed the enzymatic assay results that
 TS was responsible for the formation of 3 β -tigloyloxytropine, and indicated that TS was involved in
 the biosynthesis of calystegine A3, the major compound of calystegines. Please rephrase sentences, as
 gene silencing did not confirm the enzyme assay results. Both in planta and in vitro assays showed that
 TS catalyzes the formation of 3 β -tigloyloxytropine in the biosynthetic pathway of calystegine A3.

**Response:** Thank you for the precise and useful language editing. We directly used the sentence you
 wrote, “Both in planta and in vitro assays showed that TS catalyses the formation of
 3 β -tigloyloxytropine in the biosynthetic pathway of calystegine A3”, in the manuscript.

**14. Comment:** Line 221-230: Again quite surprising with overexpression studies as only one
 metabolite is accumulated and there is no change in downstream or competitive pathway branches.

**Response:** We reanalyzed the metabolite data obtained by LC-MS and found that the
 3 β -acetoxytropine content also increased when TS was overexpressed in hairy root cultures of *A.*
 *belladonna*. The results for 3 β -acetoxytropine were included in the revised manuscript (Figure 3).
 Overexpressing one enzyme does not eventually alter the production of metabolites in downstream or
 competitive pathway branches. For example, overexpression of CYP82M3 (catalysing the formation of
 tropinone) increased tropinone production but did not alter the production of littorine, hyoscyamine or
 scopolamine (the three tropane alkaloids are located in the downstream pathway) (Zeng *et al.* 2022).

Overexpression of PMT (catalysing the formation of N-methylputrescine as the upstream precursor in
 the biosynthesis of tropanol esters) increased the production of N-methylputrescine but did not change
 the production of tropinone, tropine, pseudotropine, calystegines, hyoscyamine or scopolamine (Rothe
 *et al.* 2023). The main reason is that rate-limiting enzymes in downstream pathways strictly govern
 metabolite flux. We explained the results in the second paragraph of the Discussion in the original
 manuscript.

**Supplementary Fig. S6. The influence of TS overexpression on the content of compounds**
 **involved in the competitive metabolic flow of TS in *A. belladonna* hairy root cultures.** (A)
 3 α -Tropanol. (B) Tropanol hexoside. (C) Littorine. (D) Hyoscyamine. (E) Scopolamine.

**15. Comment:** Figure 4A: is it N-32 or N-43?

**Response:** This term is N-32, and the typo has been corrected.

**16. Comment:** How did authors get soluble TS protein? I did not see this information in relevant
 results section?

**Response:** To obtain soluble TS, we used the pMAL-c5x plasmid to express TS and its mutants fused
 with a maltose binding protein (MBP) tag. MBP is a common protein expression tag, as it is known to
 significantly enhance the solubility of many proteins. A description of this content can be found in the
 "Purification of MBP-tagged TS and enzymatic assessment" section of the Materials and Methods.

**17. Comment:** Line 231-248: If TS has mitochondrial targeting signal, did authors remove this
 sequence for *E. coli* expression? Also authors should do localization of TS without these 32 amino
 acids in tobacco to support their claims.

**Response:** This is a wonderful point. We performed relevant experiments during the revision. When TS
 without the 32-amino-acid peptide in the TS N-terminus (TS^{Del-N32}) was expressed in *E. coli*, its
 catalytic activity decreased markedly (Supplementary Fig. S9), suggesting that the 32-amino-acid
 mitochondrial signal of TS plays a crucial role in enzymatic activity. Based on the 3D structure of the
 TS generated by AlphaFold2, its 32-amino-acid mitochondrial signal was shown to constitute the core
 scaffold.

**Supplementary Fig. S9. Effect of removing 32 amino acids at the N-terminus on the catalytic**
 **activity of TS.** (A) The removal of 32 amino acids at the N-terminus resulted in an extreme decrease in
 catalytic activity. (B) The 32 N-terminal amino acids include a β-sheet structure consisting of the TS
 core scaffold. 32 Amino acids at the N-terminus were highlighted in red. ** indicates a significant
 difference at $P < 0.01$ according to the two-sided independent sample t -test.

Furthermore, we performed subcellular localization analysis of TS without the 32-amino acid signal.
 The results indicated that TS without the 32-amino-acid signal still exhibited mitochondrial localization.
 Therefore, we speculate that the mitochondrial localization signal of TS is located at its N-terminus and
 is also distributed at other regions. However, the TS mitochondrial targeting signal from the
 non-N-terminus was not predicted by TargetP 2.0, etc., and we found no evidence for other
 mitochondrial signals in TS. Previous studies on mitochondrial localization signals reported that
 mitochondrial localization signals were located within protein sequences and were not cleaved. These
 references are listed below:

Brix, J., Rüdiger, S., Bukau, B., Schneider-Mergener, J. & Pfanner, N. Distribution of binding
 sequences for the mitochondrial import receptors Tom20, Tom22, and Tom70 in a
 presequence-carrying preprotein and a non-cleavable preprotein. *J. Biol. Chem.* 274, 16522–16530
 (1999).

Rapaport, D. Finding the right organelle. Targeting signals in mitochondrial outer-membrane proteins.
 *EMBO Rep.* 4, 948–952 (2003).

Herrmann, J. M. & Hell, K. Chopped, trapped or tacked - Protein translocation into the IMS of
 mitochondria. *Trends Biochem. Sci.* 30, 205–212 (2005).

Jensen, R. E. & Dunn, C. D. Protein import into and across the mitochondrial inner membrane: Role of
 the TIM23 and TIM22 translocons. *Biochim. Biophys. Acta - Mol. Cell Res.* 1592, 25–34 (2002).

Hell, K. The Erv1-Mia40 disulfide relay system in the intermembrane space of mitochondria. *Biochim.*
*Biophys. Acta - Mol. Cell Res.* 1783, 601–609 (2008).

The subcellular localization results of TS without 32 amino acids and the activity of the truncated TS
without the 32-amino-acid signal in its N-terminus were added to the revised manuscript
(Supplementary Fig. S8 and Supplementary Fig. S9).

**Supplementary Fig. S8 Subcellular localization analysis of TS without the 32 amino acids at the**
**N-terminus.** YFP, yellow fluorescence from YFP. Monti-Red, Monti-red fluorescence-labelled
mitochondria. Merge YFP+RED, the merged images for the yellow fluorescence and monti-red
fluorescence. Chlorophyll, chlorophyll spontaneous fluorescence. Bright, bright field image.
Overlapping images of all the channels mentioned above were merged. TS^{Del-N32}-YFP, TS without the
32 amino acids at the N-terminus fused with YFP.

Our results and those of previous publications demonstrate that the mechanism underlying
mitochondrial localization is unclear and complicated. We plan to conduct in-depth research on the TS
localization mechanism in future work.

**18. Comment:** Figure 7 can be moved to supplementary. It's quite difficult to follow all these
mutational studies.

**Response:** Figure 7 was moved to the supplementary materials (Supplementary Fig. S11). Additionally,
the orders of the relevant figures were updated in the manuscript and supplementary materials.

**19. Comment:** Line 409-419: Did authors infiltrate tigloyl-coA as a substrate after agro-infiltration? I
wonder whether *N. benthamiana* produce tigloyl CoA? I did not see this information anywhere in
manuscript. Did authors try to feed tropinone, which is extremely cheap substrate and can result in
enhancement of tigloidine instead of infiltrating 7-8 enzymes?? Authors should also include version of
TS without mitochondrial signal in benthamiana assay and test whether tigloidine is formed or not?
This will provide the information where sub-cellular localization of TS is essential for activity or not?

**Response:** *N. benthamiana* can produce tigloyl CoA. According to previous studies, tigloyl CoA is
derived from the degradation of isoleucine in plants (Basey and Woolley. 1973; Beresford and Woolley.
1974), insect (Attygalle *et al.* 2007) and human (Robinson *et al.* 1956) species. Tigloyl esters are
widely found in diverse plant metabolites, such as alkaloids and terpenoids (Kraus *et al.* 1987; Okada
*et al.* 2005; Aarthy *et al.* 2018). With respect to tobacco metabolites, tigloyl esters were also detected
(Mookherjee and Wilson. 1990). These references are listed below:

Basey, K. & Woolley, J. G. Biosynthesis of the tigloyl esters of Datura: Cis-trans isomerism.
*Phytochemistry* 12, 2883–2886 (1973).

Beresford, P. J. & Woolley, J. G. Biosynthesis of ticloidine in *Physalis peruviana*. *Phytochemistry* 13,
2143–2144 (1974).

Attygalle, A. B., Wu, X. & Will, K. W. Biosynthesis of tiglic, ethacrylic, and 2-methylbutyric acids in a
carabid beetle, *Pterostichus* (*Hypherpes*) *californicus*. *J. Chem. Ecol.* 33, 963–970 (2007).

Robinson, W. G., Bachhawat, B. K. & Coon, M. J. Tiglyl coenzyme A and alpha-methylacetoacetyl
coenzyme A, intermediates in the enzymatic degradation of isoleucine. *J. Biol. Chem.* 218, 391–400
(1956).

Kraus, W. et al. Structure determination by nmr of azadirachtin and related compounds from
*azadirachta indica* a. Juss (*Meliaceae*). *Tetrahedron* 43, 2817–2830 (1987).

Okada, T., Hirai, M. Y., Suzuki, H., Yamazaki, M. & Saito, K. Molecular characterization of a novel
quinolizidine alkaloid O-tigloyltransferase: cDNA cloning, catalytic activity of recombinant protein
and expression analysis in *Lupinus* plants. *Plant Cell Physiol.* 46, 233–244 (2005).

Aarthy, T. et al. Tracing the biosynthetic origin of limonoids and their functional groups through stable
isotope labeling and inhibition in neem tree (*Azadirachta indica*) cell suspension. *BMC Plant Biol.* 18,
(2018).

Mookherjee, B. D. & Wilson, R. A. Tobacco constituents - Their importance in flavor and fragrance
chemistry. *Perfum. Flavorist* 15, 27–49 (1990).

These previous studies demonstrated that tigloyl-CoA can be produced by plants, including tobacco.
Moreover, based on the experiments in our study, tobacco provides tigloyl-CoA as an acyl donor. When
ODC, PMT, MPO, PYKS, CYP82M3, TRII and TS^{S40T} were co-expressed in tobacco leaves, tigloidine
was produced (Fig. 8). When TS was expressed in tobacco leaves fed 3 β -tropanol, tigloidine was
produced (Supplementary Fig. S13). When TS was expressed in tobacco leaves fed 3 β -tropanol and
tiglic acid, tigloidine was produced at a slightly increased level (Supplementary Fig. S13). When PcICS
and TS (or TS^{S40T}) were co-expressed in tobacco leaves fed 3 β -tropanol and tiglic acid, tigloidine was
produced at markedly increased levels (Supplementary Fig. S13). We did not use tropione when
feeding tobacco leaves because pseudotropine (3 β -tropanol) is also inexpensive. Importantly,
3 β -tropanol is a substrate of TS.

As the truncated TS without 32 amino acids in its N-terminus lost most of its activity (Response 17,
Supplementary Fig. S9), we did not use TS without a mitochondrial signal in the tobacco assay.

**Supplementary Fig. S13 Co-expressing PcICS and TS in tobacco and feeding 3 β -tropanol and**
**tiglic acid to produce 3 β -tigloyloxytropane.** A. LC-MS analysis of 3 β -tigloyloxytropane in tobacco

extracts. B. The contents of 3 β -tigloyloxypropane in tobacco extracts. ** indicates a significant
difference at P < 0.01 according to the two-sided independent sample *t*- test.

**20. Comment:** The discussion is very long and it's quite repetitive explaining the results. I suggest
reducing it significantly. What are the key messages that one needs to understand to appreciate the
context and importance of this study?

**Response:** We have reorganized and significantly shortened the Discussion section in response to your
kind suggestion. The changes are shown in the revised manuscript. Briefly, we have highlighted and
discussed the main contributions of our study on 1) understanding the biosynthetic pathway of
calystegines; 2) the unexpected mitochondrial localization of TS; 3) the catalytic mechanism of TS,
including the acceptor binding pocket and acyl donor promiscuity; 4) the advantage of our strategy to
produce 3 β -tigloyloxypropane in *E. coli*.

**Reviewer #2 (Remarks to the Author):**

**1. Comment:** In the present manuscript, Zihua Liao and co-workers reported a unusual mitochondria
localized BAHD acyltransferases (TS) from *Atropa belladonna*, which catalyzed the esterification of
3 β -tropanol to form 3 β -tigloyloxytropane specifically. The function of TS was further confirmed in
planta by gene silence and overexpression. The catalytic mechanism was discussed by molecular
docking, which enable to generate a mutant (TS^{S40T}) with higher activity. They also successfully
reconstituted the pathway in tobacco and *E. coli*, which provide a basis on 3 β -tigloyloxytropane
production via metabolic engineering. The authors should address the following minor comments and
concerns below before it could be published in Nature communications.

**Response:** We are encouraged by your comments and have addressed your professional concerns point
by point. Incidentally, we have polished the language through the Nature Research Editing Service, and
significantly reorganized the Introduction and Discussion sections in the revised manuscript, as kindly
suggested by reviewer #1.

**2. Comment:** TS showed substrate promiscuity in vitro. Did the authors see any other acylated
products when the whole pathway was reconstituted in the *Nicotiana benthamiana*, which can provide
various acyl donors in vivo.

**Response:** According to your comments, we reanalyzed the LC-MS data. 3 β -acetoxytropane was
detected in *Nicotiana benthamiana* coexpressing EnODC, AbPMT, AbMPO, AbPYKS, AbCYP82M3,
DsTRII and TS. These results were added to the revised manuscript (Figure 7).

**Fig. 7. De novo synthesis of 3β-tigloyloxytropane in *N. benthamiana*.** (A) Reconstruction of the
 3β-tigloyloxytropane biosynthetic pathway in *N. benthamiana*. (B) LC-MS detection of target
 metabolites in tobacco leaves. (C) The contents of tropinone. (D) The contents of hygrine. (E) The
 contents of 3β-tropanol. (F) The contents of 3β-tigloyloxytropane. (G) The contents of
 3β-acetoxytropane. The control represents tobacco leaves expressing YFP. TS (TS^{S40T}) represents
 tobacco leaves coexpressing six TA genes and TS (TS^{S40T}). The data are presented as means values
 +/- s.d. (n = 3). ** indicates a significant difference between the wild-type TS and its mutant group at
 P < 0.01 according to the two-sided independent sample *t*-test.

**3. Comment:** Heterologous expression system always cannot provide sufficient amount of cofactors
 for biosynthetic genes from other organisms. In *E. coli*, this problem was solved by introducing PcICS.
 The authors should also check the reactivity of PcICS in tobacco to see whether it can boost the
 production of 3β-tigloyloxytropane when tiglic acid is infiltrated.

**Response:** Yes, tobacco cells cannot provide enough tigloyl-CoA for the biosynthesis of
 3β-tigloyloxytropane. We performed the relevant experiments you suggested before submission
 (Supplementary Fig. S12). The results showed that even when PcICS was expressed and tiglic acid was
 fed into tobacco, the level of 3β-tigloyloxytropane did not significantly increase (Supplementary Fig.
 S12), probably due to the insufficient supply of 3β-tropanol. Furthermore, we performed other
 experiments. When TS was expressed in tobacco leaves fed 3β-tropanol, 3β-tigloyloxytropane was
 produced (Supplementary Fig. S13). When TS was expressed in tobacco leaves fed 3β-tropanol and
 tiglic acid, 3β-tigloyloxytropane was produced at a slightly increased level (Supplementary Fig. S13).
 When PcICS and TS (or TS^{S40T}) were coexpressed in tobacco leaves fed 3β-tropanol and tiglic acid,
 3β-tigloyloxytropane was produced at markedly increased levels (Supplementary Fig. S13). These
 results suggested that a sufficient supply of substrates through feeding facilitated the production of
 3β-tigloyloxytropane.

**Supplementary Fig. S12 Enhancing the synthesis of tigloyl-CoA did not increase the yield of**
 **3β-tigloyloxytropine in tobacco reconstructed via the biosynthetic pathway.**

**4. Comment:** The reaction mechanism in Figure 5E is not correct. The arrow pushing should begin
 from the NH at His162 and end with the protonation of -SCoA.

**Response:** Thank you for your correction. With the help of several organic synthetic chemists, we have
 redrawn the catalytic mechanism of TS. The corrected reaction mechanism is shown in Figure 5E.

**Fig. 5 Catalytic mechanism of TS.** (A) Schematic model of the catalytic pocket that contains
 3β-tropanol and tigloyl-CoA. (B) Key residues in the substrate pocket combined with 3β-tropanol. (C)
 Comparison of the relative activities of TS and TS^{H162A} using tigloyl-CoA as the acyl donor. (D)
 Comparison of the relative activity of TS and TS mutants (3β-tropanol-binding residues mutated to

alanine) using tigloyl-CoA as the acyl donor. (E) 3 β -Tropanol and acyl donors are converted to
3 β -tropanol esters via the catalysis of TS. ** indicates a significant difference between the wild-type
TS and its mutant group at P < 0.01 according to an independent sample *t* test.

**5. Comment:** The phylogenetic tree in Figure 2D and chemical structures in Figure 2E are too small. It
better to show these two in a new figure.

**Response:** According to the reviewer's suggestions, we rearranged Fig. 2 to clarify this point. The
content of Figure 2D is presented clearly in PDF format in Supplementary Data S3.

**6. Comment:** Ref3 should be Nat. Commun. 13, 4994 (2022)

**Response:** We corrected this.

**7. Comment:** Ref10 should be Stem Cell Res. Ther. 10, 178 (2019)

**Response:** We corrected this.

**8. Comment:** Ref17 should be Nat. Commun. 13, 3832 (2022)

**Response:** We corrected this.

**9. Comment:** Ref27 should be Proc. Natl. Acad. Sci. U. S. A. 119, e2215372119 (2022)

**Response:** We corrected this.

**10. Comment:** Ref31 should be Proc. Natl. Acad. Sci. U. S. A. 120, e2302448120 (2023).

**Response:** We corrected this.

**11. Comment:** Ref41 should be Nat. Commun. 12, 1563 (2021).

**Response:** We corrected this.

**12. Comment:** Ref42 should be Plant Methods 16, 149 (2020)

**Response:** We corrected this.

**13. Comment:** Ref43 should be J. Pineal Res. 63, e12429 (2017).

**Response:** We corrected this.

**14. Comment:** Ref62 should be Nat. Commun. 14, 597 (2023).

**Response:** We corrected this.

**15. Comment:** Ref63 should be Planta 252, 6 (2020).

**Response:** We corrected this.

**16. Comment:** Ref64 should be Sci.Rep. 6, 33797, (2016).

**Response:** We corrected this.

**17. Comment:** Ref69 should be BMC Bioinformatics 11, 431 (2010)

**Response:** We corrected this.

**18. Comment:** Ref77 should be Nat. Commun. 9, 5281 (2018)

**Response:** We corrected this.

REVIEWERS' COMMENTS

Reviewer #1 (Remarks to the Author):

Review report attached.

Reviewer #2 (Remarks to the Author):

The authors have thoroughly addressed my comments. I have no further feedback.

Authors have addressed all the concerns raised in the earlier review. Nice Job, congratulations !!!

The manuscript should be accepted for publication provided that authors address following minor but quite important comments. The reorganization of Figure panels is must, as authors did not follow the order of text and relevant figure panel reference.

Abstract Line26: remain unknown instead of are unknown

Abstract Line30-32: After recombinant protein statement, these transgenic results....This does not make sense. Add VIGS data sentence if authors meant to say that

Line 116: use “identify” instead of “screen”

Be consistent while using 3 β -Tigloyloxytropine, Fig. 2A its 3 β -tigloyloxytropine

Line 122: keep HMMER search, not hmmsearch

Line 124-125: could produce instead of produced

Fig. 2E: be consistent, in text authors used ‘Sly’ and in the figure its ‘Sl’

Fig. 3. Legend: Functional characterization is more appropriate than functional identification. Also, please rewrite legend for panel C-I, these are VIGS results. There is no mention of VIGS in the C-I legends.

Supplementary Figure 8 and 9: Heading of the legend should be in bold.

Fig.5 needs reorganization. 5C was mentioned before 5B, while 5E was not mentioned in the results but only in discussion. Logical flow and order need to be maintained as text goes further. Please check this throughout the manuscript. Another example, FIG. 6G was mentioned before Fig. 6D, E and F.

Line 334: remove ab initio, no need

Line 360: *Pseudomonas chlororaphis* should be in italics

Fig. 7: Why authors used DsTRII instead of Atropa TRII in reconstitution? did I miss any logical explanation for this in the previous versions?

Fig. 8: panel A, typo error. Also, mention in the legend that HS-CoA is coenzyme A (CoA). Again, 8C is not mentioned in the results, authors jumped directly to 8D

Line 410: economical, not economically. Rephrase 'researcher have shown that....'
Something like 'previous/earlier reports/studies showed that...'

Methods: is monti-red dye same as MitoTracker Red? This is confusing. In methods, I see Mitotracker Red, while in results it's Monti-red. Please clarify.

We sincerely appreciate your professional comments that substantially improve our manuscript. The
point-by-point responses are listed below. Moreover, the manuscript is revised and formatted according
to the editorial requirements.

**1. Comment:** Abstract Line26: remain unknown instead of are unknown.

**Response:** We did so. Moreover, the abstract has been shortened to 149 words according to the editorial
requirement.

**2. Comment:** Abstract Line30-32: After recombinant protein statement, these transgenic results....This
does not make sense. Add VIGS data sentence if authors meant to say that.

**Response:** The abstract has been rewritten according to the editorial requirement.

**3. Comment:** Line 116: use “identify” instead of “screen”.

**Response:** We corrected this.

**4. Comment:** Be consistent while using 3β -Tigloyloxytropene, Fig. 2A its 3β -tigloyloxytropene.

**Response:** They are unified as “ 3β -tigloyloxytropene” in the revised manuscript.

**5. Comment:** Line 122: keep HMMER search, not hmmsearch.

**Response:** We corrected this.

**6. Comment:** Line 124-125: could produce instead of produced.

**Response:** We corrected this.

**7. Comment:** Fig. 2E: be consistent, in text authors used ‘Sly’ and in the figure its ‘SI’.

**Response:** They are unified as “Sly” in the revised manuscript.

**8. Comment:** Fig. 3. Legend: Functional characterization is more appropriate than functional
identification. Also, please rewrite legend for panel C-I, these are VIGS results. There is no mention of
VIGS in the C-I legends.

**Response:** We corrected these according to your professional suggestion.

**9. Comment:** Supplementary Figure 8 and 9: Heading of the legend should be in bold.

**Response:** We corrected them.

**10. Comment:** Fig.5 needs reorganization. 5C was mentioned before 5B, while 5E was not mentioned
in the results but only in discussion. Logical flow and order need to be maintained as text goes further.
Please check this throughout the manuscript. Another example, Fig. 6G was mentioned before Fig. 6D,
E and F.

**Response:** Fig.5 and Fig.6 was reorganized according to your professional suggestion.

**11. Comment:** Line 334: remove ab initio, no need.

**Response:** We deleted them.

**12. Comment:** Line 360: *Pseudomonas chlororaphis* should be in italics

**Response:** We did so.

**13. Comment:** Fig. 7: Why authors used DsTRII instead of Atropa TRII in reconstitution? Did I miss
any logical explanation for this in the previous versions?

**Response:** Among the reported TRIIs, DsTRII from *Datura stramonium* has the highest affinity for
tropinone (<https://www.brenda-enzymes.org/enzyme.php?ecno=1.1.1.236>). The K_m value of DsTRII is
0.033 mM. And the K_m value of AbTRII is 0.09 mM. Thus, we used DsTRII in this study. Details can
be found in this reference: Dräger B. Tropinone reductases, enzymes at the branch point of tropane
alkaloid metabolism. *Phytochemistry* 67, 327-37 (2006).

**14. Comment:** Fig. 8: panel A, typo error. Also, mention in the legend that HS-CoA is coenzyme A
(CoA). Again, 8C is not mentioned in the results, authors jumped directly to 8D.

**Response:** We corrected this typo error and added annotation about HS-CoA following your
suggestion.

**15. Comment:** Line 410: economical, not economically. Rephrase ‘researcher have shown that...’
Something like ‘previous/earlier reports/studies showed that...’

**Response:** We appreciated your professional comments and revised them in the texts.

**16. Comment:** Methods: is monti-red dye same as MitoTracker Red? This is confusing. In methods, I
see Mitotracker Red, while in results it’s Monti-red. Please clarify.

**Response:** They are unified as “MitoTracker Red” in the revised manuscript.

Many thanks indeed!

Sincerely,

Zhihua Liao